# Phenotypic plasticity promotes recombination and gene clustering in periodic environments

Davorka Gulisija[1] & Joshua B. Plotkin[1]

While theory offers clear predictions for when recombination will evolve in changing environments, it is unclear what natural scenarios can generate the necessary conditions. The Red Queen hypothesis provides one such scenario, but it requires antagonistic host–parasite interactions. Here we present a novel scenario for the evolution of recombination in finite populations: the genomic storage effect due to phenotypic plasticity. Using analytic approximations and Monte-Carlo simulations, we demonstrate that balanced polymorphism and recombination evolve between a target locus that codes for a seasonally selected trait and a plasticity modifier locus that modulates the effects of target-locus alleles. Furthermore, we show that selection suppresses recombination among multiple co-modulated target loci, in the absence of epistasis among them, which produces a cluster of linked selected loci. These results provide a novel biological scenario for the evolution of recombination and supergenes.

---

[1] Department of Biology, University of Pennsylvania, Philadelphia, PA 19104, USA. Correspondence and requests for materials should be addressed to D.G. (email: davorka@sas.upenn.edu)

The evolution of genetic recombination is a subject of longstanding interest, with tremendous development as well as outstanding questions (reviewed by Otto[1]). Empirical work has shown that the recombination rate is under genetic control[2–10] and can respond to selection[11]. However, our understanding of how recombination arises comes primarily from theory. Theory suggests that recombination will evolve in populations with negative linkage disequilibrium between beneficial alleles[12–14] (negative LD, where genotypes with extreme fitness effects are underrepresented relative to expectation). In finite populations, the evolution of recombination also requires a mechanism to generate constant and considerable diversity in order to sustain LD sufficient for selection to overcome genetic drift. These conditions limit the scenarios that promote recombination in nature to (1) a steady influx of mutations in combination with Hill–Robertson interference[15–17] or (2) constantly changing biotic environments under antagonistic coevolution between species[18,19]. Here we propose a qualitatively different scenario for the evolution of recombination in finite populations subject to changing abiotic environments, called the "genomic storage effect"[20] (Fig. 1).

Models of evolution in changing environments[14] suggest that recombination should evolve if epistasis changes sign over a few generations[13]. The period of environmental oscillation and whether the recombination rate modifier is linked to the set of seasonally selected loci determine the recombination rate, whereas the strength of selection has a marginal impact[21,22]. This theoretical work was developed in the infinite-population limit, whereas genetic variation was considered unlikely to persist in a finite population at a seasonally selected locus[23–25]. Mechanistic scenarios of abiotic variation that produce fluctuating LD and thus promote recombination in finite populations remain unexplored.

On the other hand, changing biotic environments caused by coevolution of antagonistic species, such as parasite and host or predator and prey, provide a natural scenario for the evolution of recombination[18], known as the Red Queen hypothesis for the evolution of sex[19]. The Red Queen mechanism simultaneously produces: (1) diversity at both of two selected loci, due to negative frequency dependence from coevolution with the antagonistic species, and (2) fluctuating LD, as novel rare haplotypes become advantageous and common ones become detrimental. However, models of the Red Queen mechanism assume that the two expressed loci both contribute to the interaction between competing species, whereas, in nature, parasites may evolve mechanisms to express a single antigen (i.e., allele) at a time[26–28]. Moreover, Otto and Nuismer[29] showed that Red Queen mechanism is unlikely to explain omnipresence of sex, as recombination evolves under a rather narrow parameter range in such coevolutionary models.

The genomic storage effect[20], by contrast, operates in the absence of coevolution with an antagonistic species and does not require a steady influx of mutation, and thus it may provide a new biological scenario for the evolution of recombination. The basic idea behind genomic storage is that, when abiotic environments change periodically, alleles can survive periods of adversity by escaping (recombining) to a genetic background that ameliorates the effects of selection, e.g., a modifier background that confers phenotypic plasticity, thereby storing diversity until conditions change. Not only does genomic storage generate diversity in finite populations, at both the target and modifier loci[20], but it also generates fluctuations in the sign of LD between the two recombining loci (Supplementary Fig. 1). Thus genomic storage provides a novel biological scenario that can produce persistent fluctuating LD—a well-known mechanism for the evolution of recombination[13,14,21,22]. Previous work demonstrated that genomic storage produces balanced polymorphism[20] assuming a fixed, non-zero rate of recombination between selected loci. In this study, we examine whether or not recombination and balanced polymorphism can evolve simultaneously from a non-recombining population.

If phenotypic plasticity in a seasonally selected trait promotes recombination, this would provide a plausible mechanism that shapes the patterns of recombination distances across genomes of many organisms. It is widely appreciated that phenotypic plasticity can mitigate deleterious effects of selection in adverse or perturbed habitats[30–32]. Plasticity can be modulated by an epistatic modifier (review by Scheiner[33]). Mapping studies have confirmed quantitative trait loci (QTLs) that modulate phenotypic plasticity in animal and plant model organisms[34–36] that are known to experience seasonality in the wild. In Drosophila, for example, in addition to numerous plasticity QTLs[35,36], hundreds of polymorphic loci across the genome show substantial allelic frequency oscillations in response to seasonal environmental changes[37]. These empirical results reveal natural populations that satisfy the conditions for the genomic storage effect.

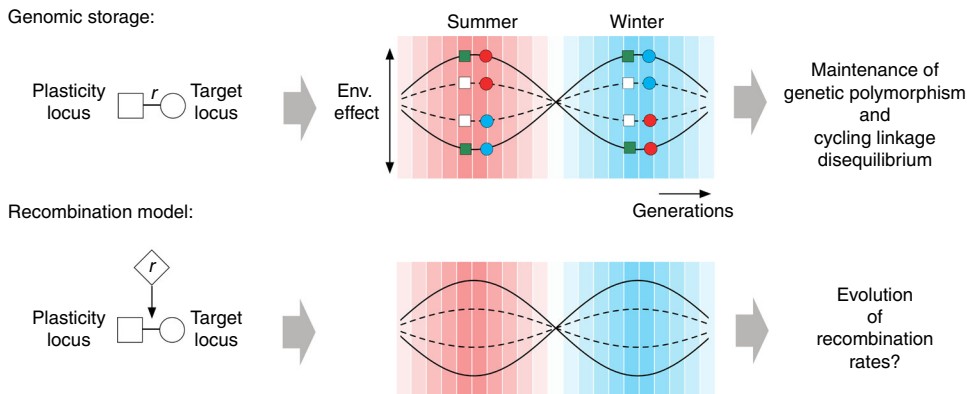

**Fig. 1** Schematic representation of the genomic storage effect and the recombination model. As environmental effects oscillate, the summer (red circles) and the winter (blue circles) alleles at the target locus recombine to a less harmful genetic background at the plasticity modifier locus. The plasticity modifier locus modulates the effect of selection at the target locus by making alleles at the target locus more (non-plastic, dark squares) or less (plastic, light squares) sensitive to selection. The dynamics between the two loci allows unfavorable alleles to be stored until conditions change, generating the genomic storage effect[20]. The genomic storage effect generates both balanced polymorphism and cycling linkage disequilibrium. Here we study whether or not this type of cycling linkage disequilibrium can lead to the evolution of recombination in initially non-recombining monomorphic finite populations

**Table 1 Common symbols and terms used in the text**

| Symbol | Description |
| --- | --- |
| $p$ | Plasticity effect—a constant that quantifies the amount of buffering of environmental effects due to the plasticity allele |
| $s_{max}$ | The maximum environmental effect, occurring at a peak season in those individuals who do not carry the plasticity allele |
| $C$ | The number of discrete generations in one oscillating cycle of periodic environmental change |
| $M$ | The plasticity-conferring allele at the modifier locus, as opposed to the non-plastic $m$ allele |
| $r_1, r_2$ | Recombination rates associated with the two alleles at the recombination modifier locus, which determines the recombination rate between the plasticity modifier locus and the target locus |
| $R$ | Recombination rate between the recombination modifier locus and the plasticity–target sequence |
| $r'$ | Recombination rate between two target loci |
| $r^*$ | The ES recombination rate, a recombination rate that cannot be displaced by any other rate in a given selection regime |

Aside from inducing the evolution of recombination between a plasticity modifier and its target locus, the genomic storage effect might also bring to proximity multiple target loci whose effects are modulated by the same plasticity locus (e.g., co-modulated due to a shared transcription factor). The clustering of loci controlling a polymorphic phenotype, i.e., the evolution of supergenes, is poorly understood (review by Thompson and Jiggins[38], and Charlesworth[39]). Charlesworth and Charlesworth[40] concluded that clustering is unlikely in the absence of initial linkage and epistasis between the selected loci, assuming the infinite-population limit. How supergenes arise in finite populations remains unclear.

In this study, we employ analytical approximations and Monte-Carlo simulations to study the evolution of the recombination rate between a plasticity modifier and its seasonally selected target locus under the genomic storage effect[20]. Unlike in prior studies, this scenario for the evolution of recombination via fluctuating LD does not depend on coevolution with antagonistic species, and it allows recombination to evolve even when a single locus expresses a trait under selection. Furthermore, we show that the genomic storage effect suppresses recombination among co-modulated loci, so that supergenes may evolve sequentially in the absence of direct frequency dependence, epistasis, and initial physical linkage between the clustering loci, in finite populations.

## Results

**Model**. To model the evolution of the recombination rate between a plasticity and target locus we generalize the Wright–Fisher population model of phenotypic plasticity described in Gulisija et al.[20] to consider three loci: a recombination modifier, a bi-allelic plasticity modifier, and bi-allelic target locus whose fitness effects depend upon a periodically changing environment. At the target locus, which codes for a periodically selected trait, an ancestral allele ($a$) is favored over the derived allele ($d$) for half of the environmental period (a season), whereas $d$ is favored over $a$ for the other half of the environmental period (as for example winter and summer alleles shown in Fig. 1). We assume a well-known and empirically supported epistatic model of plasticity[33,41], where plasticity is mediated by, for example, a transcription factor or epigenetic modifier locus. Note that we do not model the environmentally induced effects of plasticity on phenotypic trait itself, but rather, as is common in evolutionary models, we simply describe their resulting effects on fitness.

When the modifier locus carries the plasticity allele ($M$) it provides the ability to sense the environment and alter the phenotype encoded at the target locus. In particular, a cue from an adverse environment signals the plasticity allele $M$ to modify the phenotype of a detrimental allele at the target locus such that it is fitter than it would be in the absence of the plasticity allele ($m$). When conditions are not adverse and the target allele is favored, however, the plasticity allele $M$ reduces net fitness

compared to allele $m$ (due to some cost of plasticity[32,42,43]). The overall effect of such a plasticity modifier allele is thus equivalent to the action of a robustness modifier[44], such that it reduces the magnitude of periodic fitness oscillations in its carriers (broken lines in Fig. 1).

Our description of plasticity and its fitness consequences in seasonal environments does not assume that plasticity conveys either an instantaneous or, conversely, a lifelong effect on the phenotype. Rather, we simply describe the marginal fitness benefit over the lifetime of the organism. This conditionally adaptive model of plasticity has been shown[20] to promote balanced polymorphism across a wide range of parameters, including different shapes of plasticity effect, and even under stochastic environmental perturbations, when recombination is assumed to be present. Hence, we do not further explore distinct models of plasticity, but for simplicity we adopt a straightforward deterministic fitness function charactering the joint fitness effects of alleles at the plasticity locus and the target locus (see below).

We assume that the plasticity modifier locus and its target locus recombine at a rate controlled by a recombination modifier locus. The frequencies of resulting haplotypes are subject to deterministic effects of haploid selection in a constant population and of recombination between the three loci, and to the stochastic effects of genetic drift in finite populations, in each generation. In this section, we first describe the deterministic dynamics: selection and recombination, when the two competing alleles at the recombination modifier locus are present.

Combinations of alleles at the three loci (the recombination, plasticity, and target locus, with two competing alleles at each) form eight distinct haplotypes, $r_1ma$, $r_1Ma$, $r_1md$, $r_1Md$, $r_2ma$, $r_2Ma$, $r_2md$, and $r_2Md$, where $r_1$ and $r_2$ are recombination modifier alleles that produce different recombination rates between the plasticity and the target locus only. The frequencies of the eight haplotypes in each generation are first modified by selection such that post-selection frequency is the product of a haplotype's pre-selection frequency and its fitness: $x_{g,t}^{(1)} = x_{g,t} \frac{w_{g,t}}{\overline{w}_t}$, where $g = r_1ma$, $r_1Ma$, $r_1md$, $r_1Md$, $r_2ma$, $r_2Ma$, $r_2md$, or $r_2Md$, and

$$w_{r_1ma,t} = w_{r_2ma,t} = 1 - s_t, \tag{1}$$

$$w_{r_1Ma,t} = w_{r_2Ma,t} = 1 - s_t(1 - p), \tag{2}$$

$$w_{r_1md,t} = w_{r_2md,t} = 1 + s_t, \tag{3}$$

$$w_{r_1Md,t} = w_{r_2Md,t} = 1 + s_t(1 - p), \tag{4}$$

$$\overline{w}_t = \sum_g x_{g,t} w_{g,t}, \tag{5}$$

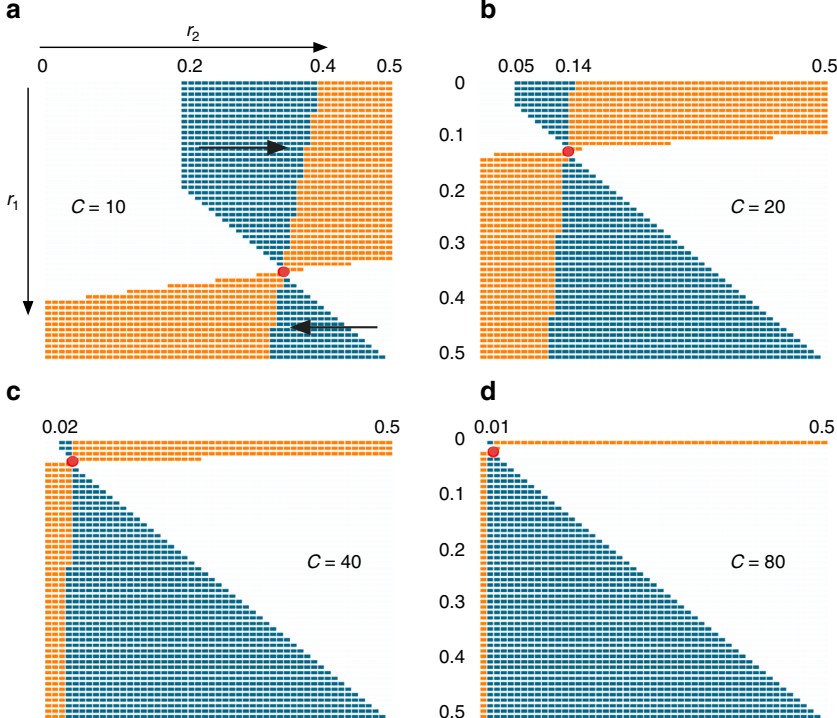

**Fig. 2** Local stability analysis of the recombination rate $r$ between the plasticity and the target locus under the genomic storage effect. For each pair of recombination rate alleles, $r_1$ and $r_2$, the corresponding color indicates whether $r_2$ fixes over $r_1$ (blue) or whether the two rates coexist at intermediate frequencies (orange), as predicted by the local stability analysis for $C = 10$ (**a**), 20 (**b**), 40 (**c**), and 80 (**d**). Empty cells imply that no plasticity–target polymorphism was present at equilibrium—i.e., no genomic storage possible. All the blue cells above the diagonal imply evolution toward higher recombination rates, while the blue cells below the diagonal imply evolution toward lower recombination rates (see arrows in the first panel). For example, under $C = 10$ an allele encoding no recombination will perish when competed with any rate ranging from 0.2 to 0.39, but it will coexist in balanced polymorphism with the rates in the range 0.4–0.5. The red dot corresponds to the ES recombination rate

with

$$s_t = s_{max} \sin\left(\frac{2\pi t}{C}\right). \qquad (6)$$

Here $s_t$ denotes the periodic environmental fitness effect at the target locus at the time $t$, which follows a sinusoidal function with maximum at $s_{max}$ over a period of $C$ discrete generations. The plasticity effect, $p$, is a constant that quantifies the reduction in the magnitude of the periodic environmental effect in those who carry the plasticity modifier allele $M$. A list of model variables is given in Table 1.

The post-selection haplotype frequencies are subsequently modified by recombination between the three loci. The physical arrangement of the three loci is assumed to be recombination–plasticity–target. The recombination modifier and the plasticity locus recombine at a fixed rate, $R$, while the plasticity and the target locus recombine according to an additive recombination phenotype between the two competing alleles at the recombination locus (see Recombination recursion in the "Methods"). Therefore, the two chromosomes recombine with the rate $r_1$ or $r_2$ if they carry the same allele, and with the rate $r_c = (r_1 + r_2)/2$ if they carry different alleles. (Note that this does not mean additive in fitness, as an intermediate phenotype might carry an advantage or disadvantage compared to the both of the recombination phenotypes.)

In the absence of genetic drift, post-recombination frequencies become the starting allele frequencies in the subsequent generation; whereas in the presence of drift the subsequent generation is formed by multinomial sampling from those frequencies. We used multinomial draws for implementing

simulations involving eight haplotypes. Whereas we used individual-based simulations for populations that included multiple different recombination-rate alleles–sampling parents (with replacement) for reproduction and recombination according to their relative fitnesses. The two simulation methods are mathematically equivalent.

We first conduct a stability analysis based on the three-locus recursion equations described above and in the "Methods", which hold in the infinite-population limit, to understand the deterministic dynamics at the three coevolving loci. Then, we study the evolution of the recombination rates, and plasticity and target alleles in finite populations, via Monte-Carlo simulations. Finally, we extend the finite-population simulations to include two or more co-modulated target loci, and an additional recombination modifier locus (or loci) that controls the recombination rates among the targets, in order to examine the evolution of target gene clustering. The details of the stability analysis and the simulation approaches are given in the "Methods".

Below we report evolution of both balanced polymorphism and recombination between the plasticity modifier locus and the target locus whose fitness effects are modulated by the modifier, in periodic environments. Moreover, if there are multiple target loci that contribute additively to fitness, we find evolution toward complete linkage among the target loci whose effects are modulated by the same plasticity modifier locus.

**Stability analysis**. As balanced polymorphism and recombination are co-dependent under the genomic storage effect, stability analysis was used not only to infer the evolutionary stable (ES)

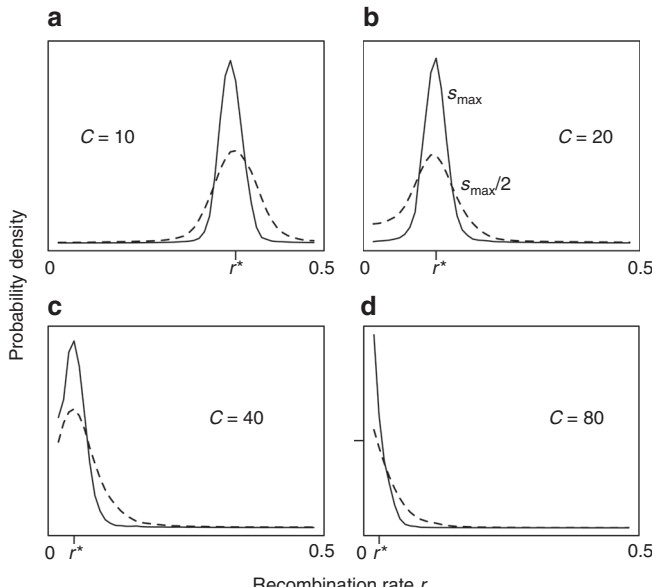

**Fig. 3** Stationary distribution of the recombination rate $r$ between the plasticity and the target locus. We assume that the recombination rate is encoded by an unlinked recombination modifier subject to recurrent mutation in a population of $N = 25,000$, with $C = 10$ and $s_{max} = 0.5$ (**a**), $C = 20$ and $s_{max} = 0.25$ (**b**), $C = 40$ and $s_{max} = 0.15$ (**c**), and 80 and $s_{max} = 0.15$ (**d**) (broken line indicates results with weaker selection, $s_{max}/2$). The tick on the horizontal axis denotes the ES recombination rate ($r^\star$) predicted by the deterministic stability analysis. $N\mu = 0.1$. One thousand replicate simulations were run for a 100 $N$ burn-in generations (where the distribution stabilized) and the stationary distribution was recorded over the next 100 $N$ generations. When the strength of selection at the target locus is reduced (by a factor of two, dotted lines), the distribution of recombination rates becomes slightly broader, but remains centered around the ES rate. The tick mark on the $y$ axis on the bottom right panel indicates height of the $y$ axis in the other panels

recombination rate, which cannot be displaced by any other rates, but also to identify the range of recombination rates that evolve with a balanced stable polymorphism at both the plasticity and target loci. This analysis predicts a range of recombination rates that we would expect to observe in populations, in the absence of a steady influx of mutations.

For all of the examined periods of environmental variation, we find evolution to a non-zero ES rate of recombination, $r^\star$, with a stable polymorphism at both the plasticity and the target locus (Fig. 2). The ES recombination rate $r^\star$ increases as the period of environmental oscillations ($C$) decreases, in accordance with earlier general models of fluctuating epistasis[14,21,22,45]. Note that the ES recombination rate we derive represents a lower bound on the potential ES rate for each periodicity, because we assumed free recombination between the recombination modifier and the plasticity–target haplotype[14,21].

Interestingly, stable polymorphic equilibria at both the plasticity and the target locus occur over a wide range of recombination rates, particularly as the environmental period increases (Fig. 2). In the absence of an allele encoding the ES rate $r^\star$, two alleles coding for a different recombination rates can coexist in proportions such that on average the population still recombines at rate roughly equal to $r^\star$, particularly with a shorter $C$.

A natural question is what is the source of selection on a recombination modifier allele? The recombination modifier allele appears selectively neutral (Eqs. 1–4 in "Model"), but it is indirectly selected due to the rate at which it produces, and finds

itself associated with, selected plasticity–target haplotypes. We can understand this indirect selection by considering the relative fitness of a recombination allele ($r_2$) to the competing recombination allele ($r_1$), over the period of fitness oscillations ($C$), which is given by

$$
\begin{aligned}
& w_{r_2}/w_{r_1} \\
&= \prod_0^C \frac{x_{r_2ma,t}w_{ma,t} + x_{r_2Ma,t}w_{Ma,t} + x_{r_2md,t}w_{dm,t} + x_{r_2Md,t}w_{Md,t}}{x_{r_1ma,t}w_{ma,t} + x_{r_1Ma,t}w_{Ma,t} + x_{r_1md,t}w_{md,t} + x_{r_1Md,t}w_{Md,t}} \\
& \times \frac{f_{r_1,t}}{f_{r_2,t}},
\end{aligned}
$$

(7)

where $f_{r_1,t}$ and $f_{r_2,t}$ are the frequencies of $r_1$ and $r_2$, in the population at time $t$. The long-term fate of a recombination allele depends on the size of the product above. $r_2$ will increase in frequency if geometric mean of the numerator exceeds that of denominator (i.e., $w_{r_2}/w_{r_1} > 1$), and it will decrease if the opposite holds[46]. Therefore, it is the relative frequencies of haplotypes in the numerator and denominator above that determine the outcome at the recombination locus. For example, consider the evolutionary dynamics over one period of fitness oscillations. Within a season (a sequence of environments of the same direction of selection, such as winter), selection promotes associations between the beneficial allele (e.g., winter allele $a$) and the non-plasticity allele ($m$), and also between the detrimental allele (summer allele $d$) and the plasticity allele ($M$), due to positive epistasis between them, $E > 0$: $a$ is relatively fitter when paired with $m$ than when paired with $M$, and $d$ is relatively fitter when paired with $M$. Thus, epistasis selects for same sign LD, i.e., $ma$ and $Md$ are overrepresented in a population relative to their expectation ($D > 0$). However, as the season changes the detrimental target allele becomes advantageous and vice versa. Now epistasis changes sign, $E < 0$: the newly detrimental allele ($a$) is fully exposed to detrimental environmental effects when paired with $m$, whereas the newly advantageous allele ($d$) is less beneficial when paired with $M$ than it would be if paired with $m$. Since it takes time for LD to change sign there is a discrepancy in the sign of epistasis and LD, such that $ED < 0$ for a period of time. In other words, at the beginning of the season LD is lagging in sign-change behind epistasis and the beneficial allelic combinations are underrepresented in the population compared to their expectation. At this point, the allele encoding a higher recombination rate will increase the proportion of fitter plasticity–target haplotypes more quickly than the one encoding a lower rate of recombination, and so the higher recombination allele will hitchhike to a higher frequency. As long as $ED < 0$, conditions will favor reduction in disequilibrium and increase in recombination. However, epistasis may eventually change the sign of LD, i.e. there will be excess of haplotypes containing allele combinations that maximize fitness, which will then favor reduction in recombination. The duration of discrepancy between the sign of epistasis and sign of linkage disequilibrium therefore determines the ES recombination rate $r^\star$, with longer periods of $ED < 0$ resulting in a higher ES recombination rate. These dynamics of cycling LD and change in epistasis are depicted in Supplementary Fig. 1.

From the description of dynamics above, it is also evident that a recombination modifier linked to the plasticity–target haplotype will gain more selective advantage than an unlinked recombination modifier, since it forms a stronger association with the fitter subpopulation.

The results of our local stability analysis, above, agree with analysis of the relative fitness of one recombination rate vs. another across each environmental cycle (Eq. 7). The ratio

$w_{r_2}/w_{r_1}$ changes in time as haplotype frequencies change during the approach to the three-locus stable attractor. In every case where we observe a fixation of rate $r_2$, as predicted by the local stability analysis (blue cells in Fig. 2), we find that the fitness of $r_2$ relative to $r_1$ ($w_{r_2}/w_{r_1}$) is uniformly greater than 1 during the approach to equilibrium. Conversely, in those cases for which the local stability analysis predicts an intermediate equilibrium frequency at the recombination locus (orange cells in Fig. 2), we indeed find that $w_{r_2}/w_{r_1} < 1$ when the frequency of $r_2$ allele exceeds the equilibrium frequency, and $w_{r_2}/w_{r_1} > 1$ when the frequency of $r_2$ is lower than equilibrium frequency. These observations of relative fitness values agree with the local stability analysis and suggest that the locally stable equilibria are globally stable as well.

**Evolution of recombination in finite populations.** For finite non-recombining populations that include genetic drift, both balanced polymorphism and recombination between a plasticity modifier and its target locus arise across a wide range of environmental periodicities (Fig. 3). We find that the stationary distribution of the recombination rate coded by an unlinked modifier allele subject to recurrent mutation is centered around the ES recombination rate predicted by stability analysis in the infinite-population limit. Also, the peaks of the stationary distribution of the recombination rate at a given periodicity are very close to each other even when the strength of selection at the target locus is varied considerably ($s_{max}$ vs. $s_{max}/2$), as expected[14,21]. However, in finite populations, especially when selection is weak, the stationary distribution of recombination rates can be very wide, such that there is a substantial chance to find a population far from the ES rate predicted by the infinite-population analysis. In fact, in some regimes where an infinite-population analysis predicts a positive ES recombination rate, selection for recombination in a corresponding finite population can be too weak to overcome drift, even due to loss of balanced polymorphism (e.g., $C = 80$, $s_{max} = 0.3$, and $N\mu = 0.01$).

Although recombination always evolves when environmental periods are short, the stationary distributions for long environmental periods ($C \geq 40$) include significant probability mass near non-recombinant modifier values (i.e., $r \sim 0$), even when balanced polymorphism is present. Nonetheless, our results on stationary rate distributions are conservative lower bounds on the evolution of recombination rate $r$, because all simulations assumed an unlinked recombination modifier ($R = 0.5$). Additional simulations showed larger equilibrium rates when the recombination modifier is flanked by the plasticity and target locus. In fact, when the recombination modifier is closely linked to the plasticity modifier ($R = 0.01$), the plasticity and target locus evolve towards free recombination ($\sim 0.5$) with $C = 10$, and recombination evolves even when environmental periods are very long ($C = 80$) (see also Supplementary Fig. 2).

Individual-based simulations reported above assumed relatively strong selection at the target locus in order to reduce computational cost. This regime of strong seasonal frequency oscillations has indeed been reported in empirical studies[47–49], even at numerous loci simultaneously[37]. Nonetheless, the genomic storage effect extends to weaker selection than shown in the figures above, provided $Ns_{max}$ is large enough[20]. To verify that recombination also evolves under weaker selection we conducted a simulation with eight haplotypes in larger populations. In these simulations, the ES recombination rate estimated by infinite-population analysis was introduced into an initially non-recombinant population. These simulations show not only that the recombination can evolve readily in large populations with the relatively small $s_{max}$ (Supplementary Fig. 3, compare left

and middle panel), but even more so if the recombination modifier is linked to the plasticity–target sequence. Conveniently, $s_{max}$ has little influence on the equilibrium recombination rate. And so the results of our individual-based simulations likely apply to wider set of selective pressures than those that can be feasibly examined by computation.

All the results above assume plasticity effect $p = 1$. But will recombination arise for weaker plasticity effects? In the deterministic limit of an infinite population, cycling LD is guaranteed for any $p > 0$. In finite populations, however, drift can disrupt diversity, especially when $Ns_{max}$ is small. Nonetheless, Supplementary Fig. 3 shows that recombination readily evolves across a wide range of plasticity effect sizes ($p \geq 0.25$) in populations of size $N = 10^6$ with $s_{max} = 0.05$ and $N\mu = 0.1$, for example.

**Evolution of clustering between two co-modulated target loci.** Genomic storage also leads to reduced recombination, and eventually complete linkage ($r' \sim 0$), among multiple, non-epistatic target loci co-modulated by the same plasticity modifier, in finite populations (Fig. 4). Hence, while genomic storage increases the recombination between the plasticity modifier and its target loci, storage has the opposite effect on the recombination rate among co-regulated target loci.

Irrespective of whether the two target loci are introduced sequentially or simultaneously, and irrespective of the initial position of the newly introduced target locus or of the location of their recombination modifiers, we find evolution of reduced recombination rates $r'$ among the target loci (i.e. clusters), positive linkage disequilibrium between the loci, higher levels of diversity (provided selection is not too strong), and increased

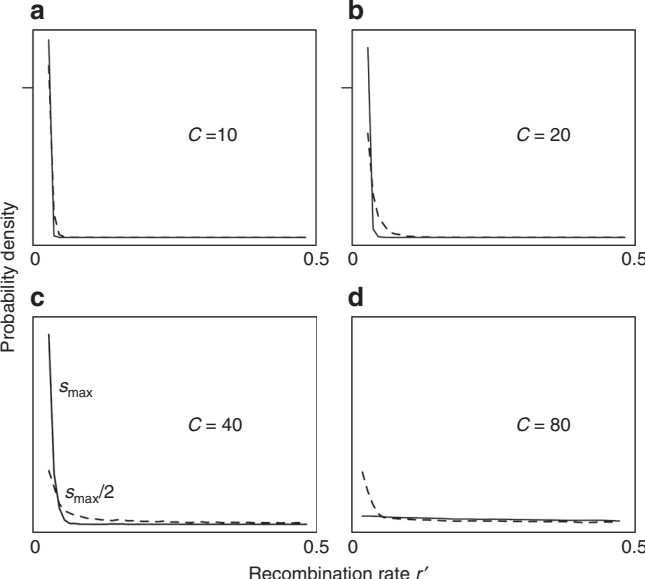

**Fig. 4** Stationary distribution of the recombination rates between two co-modulated target loci. We assume that the initial target locus starts at the optimal distance to the plasticity modifier loci and a new target locus is introduced downstream away from the plasticity modifier-target haplotype. $N = 25,000$ and $C = 10$ with $s_{max} = 0.5$ (**a**), $C = 20$ with $s_{max} = 0.25$ (**b**), $C = 40$ and $80$ with $s_{max} = 0.15$ (**c**, **d**) for each of the target loci (broken line indicates $s_{max}/2$), with $N\mu = 0.1$. One thousand replicate simulations were run for a $100 N$ burn-in generations (where the distribution stabilized) and the stationary distribution was recorded over the next $100 N$ generations (to a 2 decimal place precision). The tick marks $y$ axes in top panels indicate height of $y$ axes in bottom panels

magnitude of fitness oscillations at each locus compared to a single-target case. Interestingly, when the first target locus is controlled by a linked recombination modifier ($R = 0.01$) and the second target locus by an unlinked recombination locus, both loci cluster to the recombination distance with the plasticity locus that is favored under the linked recombination modifier. This occurs because positive LD between the two target loci, which is generated by genomic storage and genetic drift, leads to the clustering between two target loci despite the fact that the two target loci should gravitate to different recombination distances from the plasticity modifier. The clustering of target loci results in a stationary distribution of $r'$ that is almost indistinguishable from that when loci are introduced sequentially, where they gravitate to the same recombination distance from the plasticity modifier (Supplementary Fig. 4). Clustering of target loci was not observed with $C = 80$ and $s_{max} = 0.15$ (although $N\mu = 0.1$) because in that regime strong selection at clusters of ancestral–ancestral ($a$–$a$) or derived–derived ($d$–$d$) alleles pushes alleles to the boundaries and removes balanced polymorphism at the target loci–a result that occurs only in finite populations and is not predicted by a deterministic analysis.

As noted above, genomic storage generates balanced polymorphism and positive LD among multiple target loci, in the absence of epistasis between them. This result is a converse, of sorts, to Hill–Robertson interference[15] in finite populations, which produces negative LD among non-epistatic loci under non-balancing selection. Positive LD arises in our model because balancing selection maintains clusters of loci with the largest fitness variance (ancestral–ancestral or derived–derived), while mismatched haplotypes perish by genetic drift. Therefore, we find an excess of $a$–$a$ or $d$–$d$ haplotypes compared to expectation—that is, positive LD. Hence, genomic storage presents a naturally plausible mechanism for clustering of seasonal alleles, as a consequence of phenotypic plasticity in changing environments.

**Evolution of supergenes**. Clustering of multiple target loci does not arise as readily in finite populations as clustering of just two target loci. This effect arises because the relative per-locus contribution to phenotypic variance decreases as the number of target loci increases, and each such locus becomes "less visible" to selection in the presence of genetic drift. In particular, using the same parameters as in the previous section, we find that clustering is unlikely to evolve de novo among multiple co-regulated loci. However, if the target loci are initiated in tight linkage, a sequence of aligned alleles acting in unison will arise despite mutation for recombination among the target loci. Therefore, supergenes appear unlikely to evolve from polymorphic loci in the absence of their initial proximity; but genomic storage can maintain the proximity of gene complexes generated through tandem duplication and functional divergence[50–52].

Another method to evolve supergenes under genomic storage is by the sequential introduction of target loci, where each novel polymorphic locus is introduced after an initial cluster is already formed. Under this sequential model, we recover the evolution of three-loci clusters and with similar stationary distributions of the recombination rates among co-regulated target loci as found in the two-locus model. For a limited set of parameters (with $C = 20$), we find that the supergenes subsequently grow, through sequential clustering of the extant supergene with a novel locus and irrespective of its initial recombination distance ($r' \sim U[0,0.5]$), to produce supergenes including as many as $n = 8$ target loci, the maximum number we examined (Supplementary Fig. 5). At least 60% of the target loci haplotypes occur in the form of co-segregating all-$a$ alleles or all-$d$ alleles. As supergenes are created and expanded, the magnitude of frequency oscillations over the

period of environmental variation increases. Very large clusters are less likely to form when selection on each target locus is strong, however. Indeed, we observe that an 8-locus supergene is more likely to arise for smaller selection pressure ($s_{max} = 0.075$ vs. $0.1$, Supplementary Fig. 5). These results suggest that supergenes can evolve from initially distant loci if mutation at the target loci is rare enough such that a novel locus becomes polymorphic only after an initial cluster is formed, and provided selection on each target is not too strong.

The sequential model above demonstrates how supergenes can evolve via recombination reduction, or even from initially unlinked loci via genomic rearrangements[38]. Empirical data already implicate genomic rearrangements in the formation of supergenes, especially in classic examples occurring in butterfly mimicry[53] or the "social chromosome" in fire ants[54]. In the sequential model we have described above, a non-recombining rate can invade even a freely recombining population, provided there is no cost to heterozygous mating. This assumption is supported by earlier work on chromosomal rearrangements that spread in populations by suppressing recombination in heterozygotes[55]. Thus our model is equivalent to a (no-cost) model of evolution of genomic rearrangements, with the exception that the recombination phenotype is additive under our model as opposed to dominant in the case of rearrangements. The addition of dominance to our model, however, would only increase selection favoring non-recombination among target loci.

## Discussion

The Red Queen hypothesis provides a plausible scenario for the evolution of recombination under changing environments, but it is limited to cases involving coevolution with an antagonistic species, such as a parasite. This study introduces a novel scenario for the evolution of recombination under changing environments that does not require antagonistically interacting species or a steady influx of mutations: the genomic storage effect due to phenotypic plasticity. This scenario may produce both balanced polymorphism and cycling linkage disequilibrium, and it provides a natural scenario that can readily select for recombination even when a single locus structurally codes the phenotype under selection. Furthermore, while genomic storage selects for recombination between the plasticity modifier locus and its target expressed loci, the same effect also selects for complete linkage of additive co-regulated loci that are adapted to the same environment—that is, genomic storage can produce supergenes.

Our results on how the ES recombination rate depends on the period of environmental oscillation and on the strength of selection are in accordance with longstanding general theory for the effects of fluctuating epistasis[13,14,21,22]. What is novel here, however, is the specific biological scenario we propose that ensures standing diversity and fluctuating epistasis, through the genomic storage effect involving a plasticity modifier locus and seasonally selected target allele. This scenario is qualitatively different from a pair of symmetric sites, as is assumed in prior mathematical treatments of fluctuating epistasis. Moreover, we have shown that genomic storage generates the evolution of fluctuating LD and sustained polymorphism even in the presence of genetic drift.

Previous work demonstrated that genomic storage promotes balanced polymorphism across a range of parameters[20], including the variation in the strength of benefit or cost of plasticity, the period of environmental change, with and without recurrent mutation, and even in the presence of random environmental perturbations; but those analyses assumed a positive, constant recombination rate. Here we find that both recombination and balanced polymorphism can arise simultaneously in initially non-

recombining monomorphic finite populations subject to periodic selection, provided the population is sufficiently large or selection sufficiently strong. Large population sizes are not uncommon for many organisms subject to periodic environments, such as seasonally evolving organisms (e.g., copepods, a dominant zooplankton[56]). Moreover, empirical studies have reported large allele frequency oscillations under temporally varying selection[47–49], even at many loci simultaneously[37], corresponding to selection coefficients ranging from 5 to 50%. Given widespread evidence of conditions that engender genomic storage[20], this novel model may provide a plausible mechanism for the evolution of recombination in natural populations.

We have framed our model of genomic storage in terms of phenotypic plasticity, because it is well documented as a widespread example of genomic storage in periodic environments. However, the overall effect of such a plasticity modifier allele is equivalent to the action of a robustness modifier[44], which reduces the magnitude of periodic fitness oscillations in its carrier (see "Model" in the Results). Indeed, plastic phenotypes or invariable (robust) phenotypes both provide qualitatively similar fitness dynamics and are similar from an evolutionary standpoint. And so our work has implications beyond phenotypic plasticity to a wider class of natural scenarios involving buffering or sign epistatic effects.

The genomic storage effect simultaneously promotes recombination between a regulator (plasticity modifiers such as transcription factors) and its target locus, while suppressing recombination among multiple co-modulated target loci, producing clusters of target loci with aligned allelic effects. Previous research into reduction of recombination between polymorphic loci was attributed to frequency dependence and epistasis in infinite populations and suggested that this phenomenon would be unlikely in the absence of initial physical linkage[40]. By contrast, under the scenario of recombination reduction due to phenotypic plasticity in finite populations, the clustering of two target loci can readily arise independent of initial physical linkage between the loci, even in the absence of epistasis between them. The two target loci gravitate to a same recombination distance from the plasticity locus that modulates their effects. Furthermore, we find that genomic storage in finite populations promotes positive linkage disequilibrium between co-modulated loci despite the lack of epistatic interactions between them, which further promotes reduced recombination among target loci.

While two target loci under the genomic storage will evolve to be clustered independent of initial linkage, the evolution of more than two loci clustering only occurs in the presence of initial linkage between the loci or in a sequential fashion: when existing polymorphic loci are clustered, a newly polymorphic locus evolves to be tightly linked as well, such that a supergene can emerge gradually. This sequential model implies that supergenes might arise even via genomic rearrangements from initially unlinked loci provided the mutation rate at the target loci is low enough that clusters are formed before a new, unlinked, polymorphic locus arises. As supergenes emerge, selection generates strong positive linkage disequilibrium within a cluster. This joint effect of genetic drift and genomic storage on the creation of supergenes suggests that other forms of storage effects might also favor supergenes, such as storage due to population subdivision[57].

Our study highlights a role for recombination modification in the maintenance of genomic variation in seasonal environments. Polymorphism under the genomic storage effect persists only in the presence of recombination between the plasticity modifier locus and its target locus or a cluster of target loci. Such recombination will naturally evolve, we have shown, and then subsequently may promote balanced polymorphism at all of the loci. Thus, the maintenance of diversity by genomic storage is

tightly linked to evolution of recombination rates. As balanced polymorphism facilitates rapid adaptation[58–62], here recombination modification, both regulator-target distance and target loci clustering, emerges as a mechanism underlying persistence in changing environments.

While our study has focused on the effects of plasticity in a selected trait on the evolution of recombination, the effects of plasticity in recombination itself have been explored elsewhere in the context of condition-dependent recombination[63–65] (reviewed by Ram and Hadany[66]). These models show that plastic recombination may arise if it allows a recombination allele to escape to a beneficial genetic background in poor conditions. Such plastic recombination might be advantageous and broaden overall conditions for evolution of recombination in changing environments[67].

This study highlights the importance of phenotypic plasticity in shaping recombination rates across the genome, and the diverse effects it has on genetic architecture in periodic environments. We predict specific patterns of genomic distances in seasonally evolving organisms under genomic storage: transcription factors or epigenetic modifiers that modify phenotype and fitness are predicted to be unlinked or loosely linked to their target loci, depending on the periodicity of the environment to which they respond; while co-regulated expressed clusters of targets are expected to be tightly linked. The specificity of these predictions, which are unique to the genomic storage scenario, will allow this theory of recombination evolution to be compared against empirical data. More generally, the predictions of the genomic storage model can help inform empirical research on oscillating alleles and their relationship to phenotypic plasticity, just when we are rapidly uncovering many seasonal alleles[37] and plasticity modifiers[35,36] in natural populations.

## Methods

**Recombination recursion.** The eight haplotype frequencies following recombination are:

$$
\begin{aligned}
x^{(2)}_{r_1ma,t} =\, & x^{(1)}_{r_1ma,t}\Big(1 - x^{(1)}_{r_2md,t} - x^{(1)}_{r_2Md,t} - r_1 x^{(1)}_{r_1Md,t} - R x^{(1)}_{r_2Ma,t} \\
& + (1 - R - r_c + 2Rr_c)x^{(1)}_{r_2md,t} + (1-r_c)(1-R)x^{(1)}_{r_2Md,t}\Big) \\
& + x^{(1)}_{r_1Ma,t}\Big(r_1 x^{(1)}_{r_1md,t} + R x^{(1)}_{r_2ma,t} + Rr_c x^{(1)}_{r_2md,t}\Big) \\
& + x^{(1)}_{r_1md,t}\Big((R + r_c - 2Rr_c)x^{(1)}_{r_2ma,t} + (1-R)r_c x^{(1)}_{r_2Ma,t}\Big) \\
& + R(1 - r_c)x^{(1)}_{r_1Md,t}x^{(1)}_{r_2ma,t},
\end{aligned}
$$

$$
\begin{aligned}
x^{(2)}_{r_1Ma,t} =\, & x^{(1)}_{r_1Ma,t}\Big(1 - x^{(1)}_{r_2md,t} - x^{(1)}_{r_2Md,t} - r_1 x^{(1)}_{r_1md,t} - R x^{(1)}_{r_2ma,t} \\
& + (1 - R)(1-r_c)x^{(1)}_{r_2md,t} + (1 - R - r_c + 2Rr_c)x^{(1)}_{r_2Md,t}\Big) \\
& + x^{(1)}_{r_1ma,t}\Big(r_1 x^{(1)}_{r_1Md,t} + R x^{(1)}_{r_2Ma,t} + Rr_c x^{(1)}_{r_2Md,t}\Big) \\
& + R(1 - r_c)x^{(1)}_{r_1md,t}x^{(1)}_{r_2Ma,t} + x^{(1)}_{r_1Md,t} \\
& \Big((1-R)r_c x^{(1)}_{r_2ma,t} + (R + r_c - 2Rr_c)x^{(1)}_{r_2Ma,t}\Big),
\end{aligned}
$$

$$
\begin{aligned}
x^{(2)}_{r_1md,t} =\, & x^{(1)}_{r_1md,t}\Big(1 - R x^{(1)}_{r_2Md,t} - x^{(1)}_{r_2ma,t} + (Rr_c - R - r_c)x^{(1)}_{r_2Ma,t} \\
& + (1 - R - r_c - 2Rr_c)x^{(1)}_{r_2ma,t} - r_1 x^{(1)}_{r_1Ma,t}\Big) \\
& + x^{(1)}_{r_1ma,t}\Big(r_1 x^{(1)}_{r_1Md,t} + (R + r_c - 2Rr_c)x^{(1)}_{r_2md,t} + (1 - R)r_c x^{(1)}_{r_2Md,t}\Big) \\
& + R(1 - r_c)x^{(1)}_{r_1Ma,t}x^{(1)}_{r_2md,t} \\
& + R x^{(1)}_{r_1Md,t}\Big(r_c x^{(1)}_{r_2ma,t} + x^{(1)}_{r_2md,t}\Big),
\end{aligned}
$$

$$
\begin{aligned}
x_{r_1Md,t}^{(2)} =\; & x_{r_1Md,t}^{(1)}\Big(1 - r_1 x_{r_1ma,t}^{(1)} - (Rr_c - R - r_c)x_{r_2ma,t}^{(1)} \\
& - (2Rr_c - R - r_c)x_{r_2Ma,t}^{(1)} - Rx_{r_2md,t}^{(1)}\Big) \\
& + R(1 - r_c)x_{r_1ma,t}^{(1)}x_{r_2Md,t}^{(1)} + x_{r_1Ma,t}^{(1)} \\
& \Big(r_1 x_{r_1md,t}^{(1)} + (1 - R)r_c x_{r_2md,t}^{(1)} + (R + r_c - 2Rr_c)x_{r_2Md,t}^{(1)}\Big) \\
& + Rx_{r_1md,t}^{(1)}\Big(r_c x_{r_2Ma,t}^{(1)} + x_{r_2Md,t}^{(1)}\Big),
\end{aligned}
$$

$$
\begin{aligned}
x_{r_2ma,t}^{(2)} =\; & x_{r_2ma,t}^{(1)}\Big(1 - Rx_{r_1Ma,t}^{(1)} - (R + r_c - Rr_c)x_{r_1Md,t}^{(1)} \\
& - r_2 x_{r_2Md,t}^{(1)} - (2Rr_c - R - r_c)x_{r_1md,t}^{(1)}\Big) \\
& + x_{r_1ma,t}^{(1)}\Big(Rx_{r_2Ma,t}^{(1)} + (R + r_c - 2Rr_c)x_{r_2md,t}^{(1)} + R(1 - r_c)x_{r_2Md,t}^{(1)}\Big) \\
& + (1 - R)r_c x_{r_1Ma,t}^{(1)}x_{r_2md,t}^{(1)} \\
& + Rr_c x_{r_1md,t}^{(1)}x_{r_2Ma,t}^{(1)} + r_2 x_{r_2Ma,t}^{(1)}x_{r_2md,t}^{(1)},
\end{aligned}
$$

$$
\begin{aligned}
x_{r_2Ma,t}^{(2)} =\; & x_{r_2Ma,t}^{(1)}\Big(1 - Rx_{r_1ma,t}^{(1)} - (Rr_c - R - r_c)x_{r_1md,t}^{(1)} \\
& - (2Rr_c - R - r_c)x_{r_1Md,t}^{(1)} - r_2 x_{r_2md,t}^{(1)}\Big) \\
& + (1 - R)r_c x_{r_1ma,t}^{(1)}x_{r_2Md,t}^{(1)} + x_{r_1Ma,t}^{(1)} \\
& \Big(Rx_{r_2ma,t}^{(1)} + R(1 - r_c)x_{r_2md,t}^{(1)} + (R + r_c - 2Rr_c)x_{r_2Md,t}^{(1)}\Big) \\
& + x_{r_2ma,t}^{(1)}\Big(Rr_c x_{r_1Md,t}^{(1)} + r_2 x_{r_2Md,t}^{(1)}\Big),
\end{aligned}
$$

$$
\begin{aligned}
x_{r_2md,t}^{(2)} =\; & x_{r_2md,t}^{(1)}\Big(1 - (2Rr_c - R - r_c)x_{r_1ma,t}^{(1)} \\
& - (Rr_c - R - r_c)x_{r_1Ma,t}^{(1)} - Rx_{r_1Md,t}^{(1)} - r_2 x_{r_2Ma,t}^{(1)}\Big) \\
& + Rr_c x_{r_1ma,t}^{(1)}x_{r_2Md,t}^{(1)} + x_{r_1md,t}^{(1)} \\
& \Big((R + r_c - 2Rr_c)x_{r_2ma,t}^{(1)} + R(1 - r_c)x_{r_2Ma,t}^{(1)} + Rx_{r_2Md,t}^{(1)}\Big) \\
& + x_{r_2ma,t}^{(1)}\Big((1 - R)r_c x_{r_1Md,t}^{(1)} + r_2 x_{r_2Md,t}^{(1)}\Big),
\end{aligned}
$$

and

$$
\begin{aligned}
x_{r_2Md,t}^{(2)} =\; & x_{r_2Md,t}^{(1)}\Big(1 - (Rr_c - R - r_c)x_{r_1ma,t}^{(1)} \\
& - (Rr_c - R - r_c)x_{r_1Ma,t}^{(1)} - Rx_{r_1md,t}^{(1)} - r_2 x_{r_2ma,t}^{(1)}\Big) \\
& + x_{r_1Md,t}^{(1)}\Big(R(1 - r_c)x_{r_2ma,t}^{(1)} \\
& + (R + r_c - 2Rr_c)x_{r_2Ma,t}^{(1)} + Rx_{r_2md,t}^{(1)}\Big) \\
& + x_{r_2md,t}^{(1)}\Big(r_2 x_{r_2Ma,t}^{(1)} + Rr_c x_{r_1Ma,t}^{(1)}\Big) \\
& + (1 - R)r_c x_{r_1mcd,t}^{(1)}x_{r_2Ma,t}^{(1)}.
\end{aligned}
$$

**Stability analysis**. We first analyze this model in the infinite-population limit, neglecting both genetic drift and mutation. To do so, we numerically evolve the discrete-time frequency equations to identify each equilibrium that is polymorphic at both the plasticity and target locus (defined here as the case when minor allele frequency does not fall below 0.01), irrespective of the frequency of alleles at the recombination locus. We focus on such polymorphic equilibria because, in the absence of steady influx of mutation, recombination evolves only if balanced polymorphism is present at the two loci (i.e. LD is maintained by selection). Next, for each such equilibrium identified numerically we compute the Jacobean matrix of the deterministic system over a full period of fitness oscillations and the corresponding leading eigenvalue to determine if the equilibrium is locally stable or not.

We studied the deterministic dynamics across the all possible pairwise combinations of recombination rate alleles (to two-digit precision), with environmental periods $C = 10, 20, 40$, or $80$ generations. Theory[21,22] predicts that the optimum recombination rate is rather robust to the selection strength in the periodic environments, hence we do not vary the maximum environmental effect size, keeping $s_{max} = 0.1$. (But we do confirm similar results with $s_{max} = 0.01$.) Although genomic storage can occur across various strengths of the plasticity effect[20] for simplicity, we fix the parameter $p = 1$. However, later we relax this assumption for finite-population simulations (Supplementary Fig. 3). Finally, Charlesworth[14] noted that there is no optimum recombination rate for a given periodic environment, but the optimum rate increases with stronger linkage of the

recombination modifier to the selected loci ($R$). Therefore, we set $R = 0.5$ (unlinked recombination modifier) as this will produce a lower bound on the range of recombination rates that might evolve under the genomic storage, making our conclusions about the evolution of recombination conservative. Moreover, assuming $R = 0.5$ removes any potential effect of the simulated order of loci since the recombination locus freely recombines with the selected sequence. We initiate allele frequencies at the plasticity and target loci ranging from 0.05 to 0.95 (in 0.1 increments), and we examine all pairs of recombination rates, with each one given the chance to arise as the invader with initial frequency 0.01. We evolve the deterministic system for at least 1000 generation, and until either the plasticity or target locus fixes (frequency drops below $10^{-4}$) or until the same sequence (up to 8 digit accuracy) of haplotype frequencies is repeated in two consecutive environmental cycles, which we consider an equilibrium outcome.

**Monte-Carlo simulations in finite populations**. To investigate whether or not recombination modifier can invade in a non-recombining finite population, i.e., in the presence of genetic drift, and to allow recurrent mutations across the range of recombination phenotypes ($r \sim U[0, 0.5]$) and infer a realistic distribution of possible recombination rates as might occur in nature, we conducted Monte-Carlo simulations. We start each simulation in a monomorphic population at the three loci: with no recombination between the plasticity and the target locus, with allele $m$ at the plasticity modifier, and with allele $a$ at the target locus. Mutation randomly introduces diversity at each of the loci with chance $N\mu$ per generation. At the recombination modifier locus, the mutant recombination rate is randomly chosen from a uniform distribution, $U[0, 0.5]$. The other two loci reversibly mutate between alleles $m$ and $M$, or alleles $a$ and $d$. Mutation is followed by recombination and sampling with replacement (drift). Each of the two parents is randomly sampled (with replacement) and retained for reproduction proportional to its fitness relative to the maximum fitness in the population. The plasticity and the target locus recombine between the two parental chromosomes, with probability depending on the alleles they carry at the recombination modifier in an additive manner as described earlier. The recombination modifier locus recombines with the plasticity–target sequence with probability $R$. Each pair of parents is chosen from the population sequentially until the next generation of $N$ individuals is assembled. Simulations run for a burn-in duration of $100\,N$ generations (beyond the time at which genetic variance at the target and the plasticity locus and the stationary distribution at the recombination locus stabilize) and for additional $100\,N$ generations during which we record the stationary distribution of recombination rates (to two decimal place precision).

The genomic storage effect is stronger when the population size is large, when selection is strong, or when there are recurrent mutations[20]. We studied only populations of size $N = 25{,}000$ with a relative large value of $s_{max}$, corresponding to strong selection from environmental variation, since it allowed us to efficiently compute the stationary distribution in individual-based simulations. We report the stationary distribution of recombination rates between the plasticity and the target locus, $r$, under the cycle of fitness oscillations $C = 10$ with $s_{max} = 0.25$ and $0.5$, $C = 20$ with $s_{max} = 0.125$ and $0.25$, $C = 40$ and $80$ with $s_{max} = 0.15$ and $0.075$, all under $p = 1$, in an ensemble of over one thousand Monte-Carlo simulations. To demonstrate that recombination also evolves with weaker absolute selection in larger populations and with $p < 1$, we also conducted simulations based on the deterministic recursion given earlier (competing two recombination rates), but with the multinomial sampling (reproduction/drift) of haplotype frequencies. Here we examined the effects of $N = 10^5$ or $10^6$ with $s_{max} = 0.01, 0.02, 0.03$, or $0.05$, and with $p = 1$, and of $N = 10^6$ and $s_{max} = 0.5$, with $p = 0, 0.1, 0.25, 0.5$, or $1$, all with $R = 0.01, 0.1$, or $0.5$ in over 40,000 replicate simulations.

We also study a genetic system similar to the one above, but with two non-epistatic target loci, whose effects are modulated by a single plasticity modifier locus, and with a recombination locus that modifies the recombination rate between them. In this context we are especially interested in the evolution of the recombination rate among the target loci themselves. We study two particular scenarios: simultaneous evolution of the recombination rates between the plasticity and the first target locus and between the two target loci, and a sequential scenario. In the first scenario, a population migrates to the new periodic habitat and the two recombination rates are each initially monomorphic and drawn from $U[0, 0.5]$, and then subsequently evolve. In the second scenario, a population starts in equilibrium based on the three-locus model above, where the initial target locus recombines with the plasticity modifier with ES recombination rate, whereas the newly arisen polymorphic target locus has a random initial recombination rates with the existing target locus, and it is placed either between the plasticity modifier and the first target locus (for $C = 10$ or $20$) or further downstream of the first target locus ($C = 10, 20, 40$, or $80$). In both scenarios, we assume no epistasis in fitness between the multiple target loci. We postulate that if the two target loci are controlled by the same plasticity locus, then the target loci will evolve to the same recombination rate with the joint plasticity locus and thus cluster together. Additionally, in the simultaneous rate coevolution model we also examine the distance between target loci even when they would not gravitate to the same recombination to the plasticity locus, i.e., not equally distant recombination modifiers.

We obtain the distribution of recombination rates between the two target loci, $r'$, under the cycle of fitness oscillations $C = 10$ with $s_{max} = 0.25$ and $0.5$, $C = 20$ with $s_{max} = 0.125$ and $0.25$, $C = 40$ and $80$ with $s_{max} = 0.075$ and $0.15$, all under $p = 1$. The simulations run for a burn-in duration of $100\,N$ and for additional $100\,N$

generations during which we record the stationary distribution of recombination rates (to two decimal place precision).

A model with more than two target loci may exhibit qualitatively different behavior than the case of only two target loci, in finite populations. This is since polymorphism at multiple loci may interfere as it reduces the relative contribution for each locus to phenotypic variation, making each locus "less visible" to selection. We explore clustering between three target loci in two scenarios: a case where all recombination rates evolve simultaneously; and a case in which each novel polymorphic target locus is introduced sequentially at a random recombination distance ($\sim U[0, 0.5]$) to an existing cluster in equilibrium. This "sequential growth" model assumes that mutations introducing polymorphism at new locus affecting the seasonal trait are rare–so that the existing cluster reaches equilibrium before any new polymorphism arises.

For a small set of parameters (with $C = 20$), we explore the evolution of polymorphic clusters of multiple loci acting in unison (i.e., supergenes, e.g., in butterflies[53] or fire ants[54]), using the sequential growth model. Each novel polymorphic target locus is, again, introduced in a population at equilibrium where a $n$-loci cluster is already formed (with $n = 3, 4, 5, 6,$ or $7$, and, $s_{max} = 0.075$ and $0.1$). This in a sense generates a sequence of sequential two locus (super-locus and new locus) clustering. Note, that we do not observe the time till supergenes are formed as each of simulation run is conducted for $100 N$ burn-in generations beyond the time when stationary distribution is reached, and another 10,000 generations during which we record the stationary distribution of the recombination rate among target loci.

**Code availability**. Source code is available online at https://github.com/davorka/RecombinationGS.

**Data availability**. The code above was used to produce all data used in this study.

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

## Acknowledgements

This research was performed using resources and the computing assistance of the University of Wisconsin (UW)–Madison Center for High Throughput Computing (CHTC) in the Department of Computer Sciences. The CHTC is supported by UW–Madison and the Wisconsin Alumni Research Foundation and is an active member of the Open Science Grid, which is supported by the National Science Foundation and the U.S. Department of Energy's Office of Science. J.B.P. acknowledges support from the David and Lucile Packard Foundation, the U.S. Department of the Interior (D12AP00025), and the U.S. Army Research Office (W911NF-12-1-0552).

## Author contributions

D.G. proposed the idea and both authors designed the study. D.G. conducted computational work and both authors wrote the manuscript.

## Additional information

**Competing interests:** The authors declare that they have no competing financial interests.

