## [Peer Review File · Nature Communications]

Reviewers' comments:

Reviewer #1 (Remarks to the Author):

This manuscript outlines a novel model for the evolution of recombination rates between a target locus, and a plasticity locus modulating the target's fitness. The authors show that increased recombination can evolve if the fitness of the target locus fluctuates over time, but can also lead to complete linkage evolving between multiple loci targeted by the plasticity locus. It's a neat idea and seems broadly correct. I've outlined some minor suggestions for edits below.

- The model described here shares similarities with fitness-associated recombination modifier models (reviewed in Ram and Hadany 2016 "Condition-dependent sex: who does it, when and why?"), in the sense that recombination allows the plasticity allele to 'abandon' the unfit target allele following an environmental change. Some review of these models in the intro would be welcome, along with a possible discussion of similarities in the discussion if the authors feel it to be relevant.
- The terms in parentheses ('period', 'relatively fittest') on lines 82, 83 do not make sense. Please clarify them or remove.
- A notable omission from the introduction; Barton 1995 "A general model for the evolution of recombination" found that epistasis has to fluctuate every three to five generations in order for increased recombination to be selected for.
- Line 124: delete 'now'
- When introducing the model, it would be good to include a nomenclature table to aid readers. In particular: (1) Please clarify that 'p' (defined on page 10) is a constant and does not refer to an allele frequency! (2) Please clarify if r_1 , r_2 , r_c are terms mediating recombination between the plasticity and the target loci only.
- Please double-check the math typesetting on pages 11-13. There are some empty boxes where algebraic terms should be.
- Lines 262–264 with regards to stable polymorphism: is Figure S1 meant to be referenced here?
- Pages 16–17 (discussion of relative fitness of r_2 against r_1 , and measures of linkage disequilibrium): is this data shown anywhere? Does it form part of Figure S1? If not, why not measure and compare linkage disequilibrium directly between the two scenarios?
- Line 342: I take it you meant to refer to Figure S4 here, instead of Figure 4?
- Line 352: Delete 'the' before 'selection'.
- Line 379: Delete 'the' before 'genomic storage' and 'balancing selection'.
- One main conclusion is that plasticity effects can influence the evolution of increased recombination rates. Yet the model also demonstrates how complete linkage should evolve between multiple traits influenced by the same plasticity locus. Hence the discussion on whether plasticity can lead to increased recombination rates should clarify this nuanced point. That is, if I understood correctly; the model predicts that recombination can increase between plasticity alleles and co-adapted gene complexes influenced by them, but also lead to complete linkage within gene complexes.
- Better annotations of the y-axis would be desirable for Figure 2.
- Will the authors upload simulation and mathematical code online, as supplementary

material or onto another depository?

Reviewer #2 (Remarks to the Author):

Gulisija and Plotkin present a novel theoretical explanation for the evolution of recombination by examining the “genomic storage” effect due to phenotypic plasticity in periodic environments. This is an interesting situation in which to examine the impact of recombination, because essentially all organisms live in periodic environments at some scale, and phenotypic plasticity is a well-known phenomenon across all domains of life.

They find that in many situations, there is an optimal recombination distance for the plasticity locus to exist away from the target locus, and they find that a recombination modifier locus will evolve toward modifying the recombination rate to be close to the optimum under a wide variety of scenarios. I found that argument fairly convincing, although I’m not sure the parameter range is appropriate. Many of the simulations and analytical results have extremely large selection coefficients, for instance Fig 1 has a maximum selection coefficient of .1, and Fig 2 has .15 and .075, both of which are truly massive relative to my knowledge of selection coefficients estimated in modern populations. Obviously, thinking about this in the context of the evolution of recombination means that this isn’t necessarily considering evolution a modern population, but it seems unlikely to me to believe that many populations could tolerate the load induced by selection coefficients on the order of 10%. I’d like to see some more exploration of “reasonable” selection coefficients, or at least a justification of such extraordinarily large selection coefficients.

The authors also explore the evolution of highly clustered loci (“supergenes”) in this model. They find that a recombination modifier will be selected to favor reduced recombination between two target modified loci, suggesting that this can create supergenes. However, they find that in order to get larger arrays of genes, they need to be added sequentially. This seems plausible to me, as it essentially creates a sequence of 2-locus situations. I don’t recall seeing the authors point that out explicitly, but I think it’s a worthwhile point to make. Moreover, it seems that the authors don’t really explore a likely mechanism for supergene formation, which is transposition of genes, rather than recombination modification. It seems that some models in the literature for duplicated genes could at least be discussed regarding this point.

As it stands, I think the paper could be improved substantially to speak a bit more closely to some of the empirical “facts on the ground” regarding selection coefficients and proposed mechanisms, especially with regard to the formation of supergenes.

I have several additional minor comments that I think can improve the manuscript:

1) Line 43: “genetic basis for recombination”. I believe the authors mean “heritable variation in recombination rate”, as recombination obviously has a genetic basis

2) I'm not super familiar with the literature on evolution of recombination rate, so I might be mistaken here, but to me "negative linkage disequilibrium" is has a statistical meaning: that the loci are associated less frequently than expected by chance. However the authors use it to mean that the alleles have opposite fitness effects (which would certainly generate negative LD as I understand it). Maybe a bit of clarity here could be helpful to readers.

3) Line 63-66: the authors seem to imply that you only get Hill-Robertson interference in finite populations, but my understanding is that you see it even in deterministic settings. For instance, Hill-Robertson interference is common in experimental evolution studies in yeast and bacteria, where population sizes are effectively infinite.

4) Line 77: the authors make it sound like the requirement of "steady influx of mutations" is a limitation for alternative models of the evolution of recombination. However, it seems to me that the requirement of "steady influx of mutations" is a requirement for evolution, full stop. So it doesn't seem that it's too much for a theory of recombination to require!

5) Line 145: i.e. should be e.g.

6) It's not clear that any of the equations in the main text are super useful, considering they are just setting up the recursions to be solved and particularly the recursions for the 3 locus case are very long. Consider moving to methods/supplement.

7) The argument starting line 268 seems very important and I feel like could be substantially tightened up. It seems like there is some relationship to the classical theory of evolution in fluctuating environments in which the geometric mean fitness is the important quantity, and those connections could be emphasized.

8) Figure 2: am I correct to think that the point on the diagonal where everything meets is the optimal recombination rate? It might be good to indicate that directly (e.g. with a star or something) rather than simply putting the range of the optimal rate as parentheses.

I prefer to sign my reviews. My name is Joshua Schraiber.

Reviewer #3 (Remarks to the Author):

In this paper, the authors present results from deterministic and stochastic models demonstrating that in fluctuating environments recombination rates can be under selection due to a 'genome storage' effect that is based on phenotypic plasticity. I think this is a sound, interesting and generally well-written paper.

My main concern with this paper is its lack of novelty. The genome storage effect itself (fluctuating allele frequencies at a selected locus and another locus modifying the fitness effects of the first locus, published by the same authors last year) is certainly interesting. However, it seems to me that since phenotypic plasticity is only implicit, the model arrived at by the authors is a fairly generic model of fluctuating selection that also generates fluctuating epistasis and LD. Such models have been studied for a long time and in

considerable depth in the context of recombination rate evolution: some of the classic earlier work on this topic is cited (Charlesworth 1976, Sasaki & Iwasa 1987), but others is not but should be cited (especially Barton 1995, Genet Res 65:123; Gandon & Otto 2007, Genetics 175: 1835). Most of the results in the present ms merely reproduce these earlier results, e.g. that intermediate recombination rates are expected to be evolutionarily stable and that these stable recombination rates decrease with increasing periods of environmental changes. There are some novel results in the models that incorporate multiple loci and stochastic effects, but overall I felt the paper didn't add that much new to our understanding of recombination rate evolution.

A second general comment is that I think the authors should have tried to obtain general, analytical results from their deterministic model instead of relying only on numerical and simulation results. In previous work (e.g. Barton, Gandon & Otto papers given above) some ingenious techniques for analysing such models have been developed that should also be applicable to the authors' model. In addition to providing a better and more general understanding of the dynamics of recombination rate evolution, such analytical results would also enable a better comparison with previous models, including models with abiotic environmental fluctuations but also models of Red Queen dynamics.

Given these concerns, I think that although the paper should certainly be publishable in a good journal it lacks the novelty and depth of most articles that I have come across in Nat Comms.

Other, more specific comments:

-l.185: Why soft selection? To me, soft selection means local density regulation in models with structured populations, so I don't see how this is relevant here.

-l.201: It shouldn't be necessary to write down all those eight equations explicitly. Instead, this could be written as

$$x_{\{r_i, g\}}^{(1)} = x_{\{r_i, g\}} w_g / w_{\bar{g}}, \text{ with } i=1,2 \text{ and } j=ma, Ma, md, Md$$

-l.204: The fitness model is slightly odd in that it is asymmetrical: relative to the fitness of 1 attained by all genotypes at time 0, c , $2c$ etc., the disfavoured genotype is disfavoured more strongly than the favoured genotype is favoured at other times. Instead of fitness values $1-s_t$ I think it would have been more natural to assume fitness values of $1/(1+s_t)$. However, this probably doesn't make a big difference, especially for weak selection.

-l.219, "without loss of generality": It's not obvious to me why the order of loci shouldn't matter here because they all have different effects.

-l.222: Again, I don't think these equations need to be given this explicitly. They can be written using tensors, or the authors could put them into a supplementary material and/or cite earlier work (perhaps Nei 1967?). Also, in at least two instances these equations have

empty boxes instead of letters t and x, but this might also be just a pdf conversion error.

-l.226: When reading up to this point, I thought by "sampling with replacement" the authors meant random numbers were drawn from multinomial distributions with parameters given by the genotype frequencies and the population size, and then normalised to give the new frequencies. However, further down it seemed although this was indeed done for large populations, an individual-based approach was employed for most of the the stochastic simulations. I was a bit confused by this so I suggest to spell out the two different approaches more clearly and motivate them better.

- Fig. 2: It wasn't clear to me what was meant by "outcompeted rec rate" and "established rec. rate". Do you mean resident and mutation rec. rate here? Also, it would be good to have properly labeled axes on these plots.

- l. 268-293: I was surprised that there was no mentioning of epistasis here, as I would expect this to be the main driving force. Epistasis should fluctuate between positive and negative values, following the sin function in Eq. 6. (A quick calculation shows that for additive epistasis these fluctuations occur between values of $-2ps_{\max}$ and $2ps_{\max}$, but multiplicative epistasis is probably more appropriate.) Epistasis produces selection for LD of the same sign, but because LD will somewhat lag behind there will be time periods when E and D have the opposite sign, and this is when selection for increased recombination is expected. In total, the "optimal" recombination rate depends on how much time the population spends in those time period whereas those where $ED > 0$ and where there is selection against recombination. I think if this is indeed a valid explanation, it would be good to mention it and perhaps also show plots similar to Figure 6 in Gandon & Otto 2007 (Genetics 175: 1835).

-l.252-253: Instead of the imprecise phrase "optimal recombination rate r^* , that is relatively more fit than all other rates" I would suggest to use to technical term "evolutionarily stable (ES) recombination rate" here and throughout the ms.

The language could be polished throughout the manuscript to reduce grammatical errors as well as incidences of imprecise writing. Some examples:

-l.43: Imprecise - perhaps better something like "Different lines of evidence indicate a genetic basis for recombination rates, including ..."

-l.122: "moreover"

-l.137: "the" missing before "result"

- l.754: "a non-recombining rate will be outcompeted" is quite imprecise. (It's not the rate that is recombining, and it's alleles that are outcompeted.)

- l.281: "disequilibrium"

- l.281: "such to increase their fitness" sounds odd to me.

- l.328: Not sure "equilibrium" is a word, and if it is there is a "u" missing.

Point-by-point response to reviews:

Reviewer #1 (Remarks to the Author):

This manuscript outlines a novel model for the evolution of recombination rates between a target locus, and a plasticity locus modulating the target's fitness. The authors show that increased recombination can evolve if the fitness of the target locus fluctuates over time, but can also lead to complete linkage evolving between multiple loci targeted by the plasticity locus. It's a neat idea and seems broadly correct. I've outlined some minor suggestions for edits below.

Thank you for the overall assessment of the study, and for your detailed suggestions to clarify and improve the work.

- The model described here shares similarities with fitness-associated recombination modifier models (reviewed in Ram and Hadany 2016 “Condition-dependent sex: who does it, when and why?”), in the sense that recombination allows the plasticity allele to ‘abandon’ the unfit target allele following an environmental change. Some review of these models in the intro would be welcome, along with a possible discussion of similarities in the discussion if the authors feel it to be relevant.

We now review the literature on condition-dependent recombination in the Discussion (lines 560 – 568), and we describe its relationship to our own work.

- The terms in parentheses (‘period’, ‘relatively fittest’) on lines 82, 83 do not make sense. Please clarify them or remove.

Corrected.

- A notable omission from the introduction; Barton 1995 “A general model for the evolution of recombination” found that epistasis has to fluctuate every three to five generations in order for increased recombination to be selected for.

Thank you, we agree. We have added Barton 1995 and another key reference to prior mathematical studies of recombination rates under fluctuating epistasis (lines 80 – 83).

- Line 124: delete ‘now’

Corrected.

- When introducing the model, it would be good to include a nomenclature table to aid readers. In particular: (1) Please clarify that 'p' (defined on page 10) is a constant and does not refer to an allele frequency! (2) Please clarify if r_1 , r_2 , r_c are terms mediating recombination between the plasticity and the target loci only.

We have introduced a table to summarize all the variable names and symbols used in our model (Table 1).

- Please double-check the math typesetting on pages 11-13. There are some empty boxes where algebraic terms should be.

The typesetting has been corrected and moved to the Supplementary Methods.

- Lines 262–264 with regards to stable polymorphism: is Figure S1 meant to be referenced here?

We referenced Figure 2 as it demonstrates the range of recombination rates compatible with stable polymorphic equilibrium at the plasticity and at the target locus.

- Pages 16–17 (discussion of relative fitness of r_2 against r_1 , and measures of linkage disequilibrium): is this data shown anywhere? Does it form part of Figure S1? If not, why not measure and compare linkage disequilibrium directly between the two scenarios?

We now clarify that the summary of the data is given in Figure 2. We confirm selection for recombination by reporting relative fitness, because fitness is an ultimate determinant of selection. We also reference the revised Supplementary Figure 1, which now shows both cycling LD and cycling epistasis under our model.

- Line 342: I take it you meant to refer to Figure S4 here, instead of Figure 4?

Indeed, corrected.

- Line 352: Delete 'the' before 'selection'.

Corrected.

- Line 379: Delete 'the' before 'genomic storage' and 'balancing selection'.

Corrected.

- One main conclusion is that plasticity effects can influence the evolution of increased recombination rates. Yet the model also demonstrates how complete linkage should evolve between multiple traits influenced by the

same plasticity locus. Hence the discussion on whether plasticity can lead to increased recombination rates should clarify this nuanced point. That is, if I understood correctly; the model predicts that recombination can increase between plasticity alleles and co-adapted gene complexes influenced by them, but also lead to complete linkage within gene complexes.

We have now emphasized these two distinct predictions throughout the text, particularly in our discussion (lines 490 – 492 and 518 – 521).

- Better annotations of the y-axis would be desirable for Figure 2.

Corrected.

- Will the authors upload simulation and mathematical code online, as supplementary material or onto another depository?

Yes. We will make all the code publically available.

Reviewer #2 (Remarks to the Author):

Gulisija and Plotkin present a novel theoretical explanation for the evolution of recombination by examining the “genomic storage” effect due to phenotypic plasticity in periodic environments. This is an interesting situation in which to examine the impact of recombination, because essentially all organisms live in periodic environments at some scale, and phenotypic plasticity is a well-known phenomenon across all domains of life.

Thank you for your detailed suggestions to clarify and improve our manuscript.

They find that in many situations, there is an optimal recombination distance for the plasticity locus to exist away from the target locus, and they find that a recombination modifier locus will evolve toward modifying the recombination rate to be close to the optimum under a wide variety of scenarios. I found that argument fairly convincing, although I’m not sure the parameter range is appropriate. Many of the simulations and analytical results have extremely large selection coefficients, for instance Fig 1 has a maximum selection coefficient of .1, and Fig 2 has .15 and .075, both of which are truly massive relative to my knowledge of selection coefficients estimated in modern populations. Obviously, thinking about this in the context of the evolution of recombination means that this isn’t necessarily considering evolution a modern population, but it seems unlikely to me to believe that many populations could tolerate the load induced by selection coefficients on the order of 10%. I’d like to see some more exploration of “reasonable” selection coefficients, or at least a justification of such extraordinarily large selection coefficients.

Initially, we had the same initial intuition (which really arises from pop-gen theory) that large s would be rare in natural populations. But in fact empirical data from natural populations of *Drosophila* indicate that hundreds of loci experience strong, seasonal selection – which produces variation in allele frequency from ~ 0.1 to ~ 0.9 over the course of just six months (Bergland et al 2014, with average selection coefficients ranging between 5% and 50%). More generally, very strong temporal frequency oscillations have been reported in other empirical studies (Lynch 1987; Cain et al. 1990; Turelli et al. 2001). These empirical studies justify the parameter regime of our analysis. In addition, in the revised manuscript we emphasize that such strong selection is not required for the evolution of cycling LD and recombination under genomic storage (lines 375 – 388 and Supplementary Figures S1 and S4.), especially if the modifier is linked to the recombining sequence. Given that the choice of selection pressure has little effect on the ES recombination rate (lines 345 – 348), and that, for a subset of parameters, we demonstrate that recombination evolves under much weaker selection, we feel that taken together results outline expected evolution of recombination under a wide set of selection effects in natural populations.

The authors also explore the evolution of highly clustered loci (“supergenes”) in this model. They find that a recombination modifier will be selected to favor reduced recombination between two target modified loci, suggesting that this can create supergenes. However, they find that in order to get larger arrays of genes, they need to be added sequentially. This seems plausible to me, as it essentially creates a sequence of 2-locus situations. I don’t recall seeing the authors point that out explicitly, but I think it’s a worthwhile point to make. Moreover, it seems that the authors don’t really explore a likely mechanism for supergene formation, which is transposition of genes, rather than recombination modification. It seems that some models in the literature for duplicated genes could at least be discussed regarding this point.

We agree. In the revised manuscript we highlight that sequential increase in clusters is essentially a two-loci (super-locus plus new locus) clustering effect (lines 458 – 459). We also clarify that our results on multi-target clustering are not restricted to recombination reduction. We observe a non-recombining rate invading populations across the range of recombination rates, including freely recombining populations. This scenario includes distant alleles brought together by the means of chromosomal rearrangements or transposition, assuming there is no cost to heterozygous mating. We therefore added references to prior literature on cost-free chromosomal rearrangements that can spread in populations by recombination suppression in its carriers (Coyne et al. 1993).

As it stands, I think the paper could be improved substantially to speak a bit more closely to some of the empirical “facts on the ground” regarding selection coefficients and proposed mechanisms, especially with regard to the formation of supergenes.

We entirely agree. We have expanded our discussion section to highlight the empirical literature on strong seasonal allele fluctuations, and mechanisms of supergene formation by transposition. In addition, to make contact with ongoing empirical work, the revised Discussion section proposes specific patterns of allelic oscillation and linkage to examine in *Drosophila* populations as an empirical test of recombination evolution through the genomic storage effect.

I have several additional minor comments that I think can improve the manuscript:

1) Line 43: “genetic basis for recombination”. I believe the authors mean “heritable variation in recombination rate”, as recombination obviously has a genetic basis

This phrase has been reformulated (line 43).

2) I’m not super familiar with the literature on evolution of recombination rate, so I might be mistaken here, but to me “negative linkage disequilibrium”

is has a statistical meaning: that the loci are associated less frequently than expected by chance. However the authors use it to mean that the alleles have opposite fitness effects (which would certainly generate negative LD as I understand it). Maybe a bit of clarity here could be helpful to readers.

Thanks – your interpretation of the term is correct, and it agrees with our intended usage throughout the manuscript. We have clarified the meaning of negative LD from the first time we use the phrase (lines 47 – 49), and emphasized how negative LD arises in our model. (In fact, our model produces cycling LD that changes sign from positive to negative over the course of each seasonal oscillation.)

3) Line 63-66: the authors seem to imply that you only get Hill-Robertson interference in finite populations, but my understanding is that you see it even in deterministic settings. For instance, Hill-Robertson interference is common in experimental evolution studies in yeast and bacteria, where population sizes are effectively infinite.

Thank you. We have clarified that occurrence of negative LD due to Hill-Robertson effect is not limited by population size (line 70).

4) Line 77: the authors make it sound like the requirement of “steady influx of mutations” is a limitation for alternative models of the evolution of recombination. However, it seems to me that the requirement of “steady influx of mutations” is a requirement for evolution, full stop. So it doesn't seem that it's too much for a theory of recombination to require!

Well, we don't entirely agree. In the context of models to explain the evolution of recombination, a “steady influx of mutations” is, in fact, a very strong assumption. In particular, those models assume that mutational influx alone can support substantial diversity at each locus. By contrast, the mechanism we propose includes the co-evolution of recombination with balanced polymorphism, and so it does not require ongoing mutation to support significant standing diversity. We clarify this distinction in the revised text (line 61).

5) Line 145: i.e. should be e.g.

Corrected.

6) It's not clear that any of the equations in the main text are super useful, considering they are just setting up the recursions to be solved and particularly the recursions for the 3 locus case are very long. Consider moving to methods/supplement.

We agree, and so we have shortened the selection recursion, and moved the recombination recursion to the Supplementary Methods.

7) The argument starting line 268 seems very important and I feel like could be substantially tightened up. It seems like there is some relationship to the classical theory of evolution in fluctuating environments in which the geometric mean fitness is the important quantity, and those connections could be emphasized.

We now emphasize and elaborate on the relevance of the geometric mean (lines 300 – 302).

8) Figure 2: am I correct to think that the point on the diagonal where everything meets is the optimal recombination rate? It might be good to indicate that directly (e.g. with a star or something) rather than simply putting the range of the optimal rate as parentheses.

Yes, that's correct – we have modified the figure to make it clearer.

Reviewer #3 (Remarks to the Author):

In this paper, the authors present results from deterministic and stochastic models demonstrating that in fluctuating environments recombination rates can be under selection due to a 'genome storage' effect that is based on phenotypic plasticity. I think this is a sound, interesting and generally well-written paper.

Thank you for a careful reading of our manuscript, and for encouraging us to place it more properly in the context of the prior literature on recombination. In particular, your comments have compelled us to explain what aspects of our work are genuinely novel, and how they make specific predictions to guide future empirical investigations.

My main concern with this paper is its lack of novelty. The genome storage effect itself (fluctuating allele frequencies at a selected locus and another locus modifying the fitness effects of the first locus, published by the same authors last year) is certainly interesting. However, it seems to me that since phenotypic plasticity is only implicit, the model arrived at by the authors is a fairly generic model of fluctuating selection that also generates fluctuating epistasis and LD. Such models have been studied for a long time and in considerable depth in the context of recombination rate evolution: some of the classic earlier work on this topic is cited (Charlesworth 1976, Sasaki & Iwasa 1987), but others is not but should be cited (especially Barton 1995, *Genet Res* 65:123; Gandon & Otto 2007, *Genetics* 175: 1835). Most of the results in the present ms merely reproduce these earlier results, e.g. that intermediate recombination rates are expected to be evolutionarily stable and that these stable recombination rates decrease with increasing periods of environmental changes. There are some novel results in the models that incorporate multiple loci and stochastic effects, but overall I felt the paper didn't add that much new to our understanding of recombination rate evolution.

We agree with the referee: there is no mathematical novelty in this study. Indeed, once one assumes fluctuating epistasis between a pair of loci, in an infinite population, then our specific results (which are not even derived analytically) on the ES recombination rate agree with prior mathematical studies on the topic. We have revised the manuscript to make this point perfectly clear (see second paragraph of the Discussion).

However, we feel that this comment really misses what is novel and important about our study: we describe a qualitatively new and very plausible biological scenario, based on a plasticity modifier and its periodically selected target locus, that serves as a mechanism to produce the conditions of fluctuating epistasis and LD required for the evolution of recombination. Aside from the Red Queen scenario, which requires inter-specific interactions, prior works on single-species models for

the evolution of recombination in changing environments have typically assumed a pair of symmetric sites that remain polymorphic and experience fluctuating epistasis according to the phase of the environment. What our work provides is a specific biological scenario, with substantial empirical support, that guarantees these two conditions in finite (natural) populations, without assuming a Red Queen interaction or a constant influx of mutations.

The scenario we propose for generating fluctuation LD and sustained diversity is indeed novel. Note that this scenario involves two different types of loci: one locus that experiences seasonal selection and another locus that modulates the fitness effects of alleles at the first locus. This scenario is qualitatively different from a pair of symmetric epistatic sites, as assumed in prior mathematical treatments. For example, unlike in prior studies, one of the loci in our scenario experiences allelic oscillations at twice the rate of the other locus. And, unlike in prior models, recombination in our scenario depends on balanced polymorphism, and balanced polymorphism depends on recombination – i.e. they both co-evolve in finite populations.

Aside from a specific, biological scenario that generates fluctuating LD and balanced polymorphism, our work is one of very few studies to show that these conditions simultaneously arise in finite populations as well.

Finally, a large portion of our revised manuscript is devoted to explaining and analyzing the emergence of supergenes in finite populations. This portion of our work – showing that the genomic storage effect that favors recombination between plasticity and target loci also favors linkage among multiple non-epistatic target loci – is also novel and lacking from prior studies on the evolution of recombination.

In summary, we do not expect to convince the readership about the mathematical novelty of our work – indeed the literature on this subject is full of subtle techniques and rich results on ES recombination rates, once cycling LD is assumed. Rather, we hope to convey in the revised manuscript that the novelty lies in proposing a specific new biological scenario, with firm empirical grounding in plasticity modifiers, that produces the conditions known to favor recombination.

Finally, with respect to empirical relevance, we wish to point out that the scenario we propose based on a plasticity modifier and target locus makes specific and idiosyncratic predictions for rates of oscillation and recombination distances between different types of loci (e.g. those encoding seasonally selected traits versus those encoding epigenetic modifiers). As we proposed in the revised Discussion, these specific predictions can now be examined empirically in the hundreds of sites that exhibit strong seasonal allelic fluctuations in wild populations of *Drosophila* (Bergland et al 2014), for example.

A second general comment is that I think the authors should have tried to obtain general, analytical results from their deterministic model instead of relying only on numerical and simulation results. In previous work (e.g. Barton, Gandon & Otto papers given above) some ingenious techniques for analysing such models have been developed that should also be applicable to the authors' model. In addition to providing a better and more general understanding of the dynamics of recombination rate evolution, such analytical results would also enable a better comparison with previous models, including models with abiotic environmental fluctuations but also models of Red Queen dynamics.

While our model is similar in form to some previously explored models, an analytical treatment is likely to be more complicated because three loci (target, plasticity modifier, and recombination modifier) co-evolve; and there is no balanced polymorphism without positive recombination. That is to say, to analyze the evolution of recombination under genomic storage we cannot simply assume an equilibrium at plasticity–target loci and use the elegant arguments from previous work to investigate resulting optimal recombination rates. Furthermore, unlike most analytical treatments, we obtain a range of recombination rates compatible with balanced polymorphism, not merely a single ES rate; and we also calculate the stationary distribution of rates in finite populations. We present these results because we are interested in the range of recombination rates that might appear in natural populations. For all these results we used numerical stability analysis, in the infinite-population limit, and Monte Carlo simulations, in finite populations.

Whilst we appreciate the ingenuity of prior analytical techniques devised to study ES recombination rates under fluctuating epistasis, we do not feel that they can be easily adapted to our scenario. But more important, we believe the value of the work for the audience at *Nature Communications* will not be in the subtleties of mathematical analysis but rather in providing a plausible biological scenario for the evolution of recombination through plasticity modifiers; and making specific predictions for empirical studies on fluctuating allele dynamics.

Other, more specific comments:

-1.185: Why soft selection? To me, soft selection means local density regulation in models with structured populations, so I don't see how this is relevant here.

Here we referred to soft selection as originally defined by Wallace (1975, *Evolution* 29: 465-473) – an evolutionary outcome that depends on presence and frequency of other types in population of constant size, i.e. on relative fitness. But we agree that this term is not really necessary, and perhaps even confusing, and so we have removed it.

-1.201: It shouldn't be necessary to write down all those eight equations explicitly. Instead, this could be written as

$$x_{\{r_i,g\}}^{(1)} = x_{\{r_i,g\}} w_g / \bar{w}, \text{ with } i=1,2 \text{ and } j=ma, Ma, md, Md$$

Thanks, we have simplified the notation.

-1.204: The fitness model is slightly odd in that it is asymmetrical: relative to the fitness of 1 attained by all genotypes at time 0, c , $2c$ etc., the disfavoured genotype is disfavoured more strongly than the favoured genotype is favoured at other times. Instead of fitness values $1-s_t$ I think it would have been more natural to assume fitness values of $1/(1+s_t)$. However, this probably doesn't make a big difference, especially for weak selection.

Since it is a geometric and not arithmetic mean across the seasons that determines selective outcome, and since geometric mean is sensitive to outliers, some "arithmetic asymmetry" in fitness is necessary to assure that no allele is overall favored or disfavored. With the $1+s_t$ and $1/(1+s_t)$ scheme, the allele with $1/(1+s_t)$ numerator fitness function would actually become selected against. And so we believe the current formulation is more natural.

The $1+s_t$ vs $1-s_t$ scheme assures no allele is overall favored or disfavored (i.e. geometric mean relative fitness = 1), and so it allows us to explore evolution of recombination and clustering due to genomic storage free of effects of directional selection. We suspect that similar reasoning led to similar fitness descriptions in the earlier haploid models of temporally varying selection, e.g. Sasaki and Iwasa 1987 or Gandon and Otto 2007. In any case, previous work in storage effects suggests that choice of the specific numerator fitness makes little difference, as long as alleles are marginally close to neutral, even under strong fitness oscillations in finite populations.

-1.219, "without loss of generality": It's not obvious to me why the order of loci shouldn't matter here because they all have different effects.

The majority of our results, including those from the stability analysis, are obtained under assumption of an unlinked recombination modifier, and so the order in which plasticity-target loci are coded is irrelevant. We now highlight this in Methods, rather than in the model setup, to avoid confusion (lines 611 – 613). In those portions of our analysis where the recombination modifier is linked to the plasticity or target locus we are explicit about the specific ordering.

-1.222: Again, I don't think these equations need to be given this explicitly. They can be written using tensors, or the authors could put them into a supplementary material and/or cite earlier work (perhaps Nei 1967?). Also, in at least two instances these equations have empty boxes instead of letters t and x , but this might also be just a pdf conversion error.

The recursion is now deferred to the Supplementary Methods.

-l.226: When reading up to this point, I thought by “sampling with replacement” the authors meant random numbers were drawn from multinomial distributions with parameters given by the genotype frequencies and the population size, and then normalised to give the new frequencies. However, further down it seemed although this was indeed done for large populations, an individual-based approach was employed for most of the the stochastic simulations. I was a bit confused by this so I suggest to spell out the two different approaches more clearly and motivate them better.

The referee is correct: we used multinomial draws for simulations in very large populations that contained only 8 haplotypes, whereas we used individual-based simulations (drawing parents with replacement) in the smaller populations, including simulations with many more than 8 haplotypes. However, these two approaches are mathematically equivalent – the difference is purely a matter of implementation in a computer program (the multinomial implementation is more computationally efficient and really necessary in large populations). We have explained this distinction in implementation whilst highlighting their mathematical equivalence in the revised manuscript (lines 241 – 246).

- Fig. 2: It wasn't clear to me what was meant by “outcompeted rec rate” and “established rec. rate”. Do you mean resident and mutation rec. rate here? Also, it would be good to have properly labeled axes on these plots.

We corrected “established” and “outcompeted” to “adaptive” and “maladaptive” rates, respectively, and re-labeled the axes.

- l. 268-293: I was surprised that there was no mentioning of epistasis here, as I would expect this to be the main driving force. Epistasis should fluctuate between positive and negative values, following the sin function in Eq. 6. (A quick calculation shows that for additive epistasis these fluctuations occur between values of $-2ps_{max}$ and $2ps_{max}$, but multiplicative epistasis is probably more appropriate.) Epistasis produces selection for LD of the same sign, but because LD will somewhat lag behind there will be time periods when E and D have the opposite sign, and this is when selection for increased recombination is expected. In total, the “optimal” recombination rate depends on how much time the population spends in those time period whereas those where $ED > 0$ and where there is selection against recombination. I think if this is indeed a valid explanation, it would be good to mention it and perhaps also show plots similar to Figure 6 in Gandon & Otto 2007 (Genetics 175:1835).

Thank you, for suggesting this clarification to the text. We have modified the writing in this section and altered Supplementary Figure 1 to highlight that the sign of epistasis fluctuates, with LD changes lagging behind.

-1.252-253: Instead of the imprecise phrase “optimal recombination rate r^* , that is relatively more fit than all other rates” I would suggest to use to technical term “evolutionarily stable (ES) recombination rate” here and throughout the ms.

We now refer to ES recombination rate throughout the manuscript.

The language could be polished throughout the manuscript to reduce grammatical errors as well as incidences of imprecise writing. Some examples:

-1.43: Imprecise - perhaps better something like “Different lines of evidence indicate a genetic basis for recombination rates, including ...”

Corrected.

-1.122: “moreover”

Corrected.

-1.137: “the” missing before “result”

Corrected.

- 1.754: “a non-recombining rate will be outcompeted” is quite imprecise. (It’s not the rate that is recombining, and it’s alleles that are outcompeted.)

Corrected.

- 1.281: “disequilibriUM”

Corrected.

- 1.281: “such to increase their fitness” sounds odd to me.

Expression removed.

- 1.328: Not sure “equilibrial” is a word, and if it is there is a “u” missing.

“Equilibrial” is not quite a word, we agree. (It’s not in Merriam Webster, but it is used with some frequency in scientific texts). We have rephrased to avoid the word throughout the text.

Reviewers' comments:

Reviewer #1 (Remarks to the Author):

I'd like to thank the authors for making the requested corrections to the manuscript. The mechanisms of the 'genotype storage' effect have now been elucidated and are now much clearer to understand. I only have a few additional comments.

- Line 148: Delete 'and' after the semicolon.
- Line 290: Write 'a' shorter C, rather than 'the' shorter C.
- I like the new text explaining the interaction between epistasis E and linkage disequilibrium D. Yet I feel it could be refined somewhat. First, on line 317 it states that "LD will be selected against". Does it make sense for LD to be 'selected against'? I take it that you mean that $ED < 0$ will favour conditions to reduce it (increased recombination in this case). Second, would you consider expanding on Figure S1 to plot how time LD mismatches with epistasis, for specific parameter spaces? Including this measure would make it clearer when recombination would be favoured, which is not entirely clear from Figure S1.
- Line 325: Write "evident *that a* recombination modifier..."
- Lines 329–337: Did you plot the relative fitness values on Figure 2, as indicated by the colours? I don't think you did, but it not clear from the text as written. Please clarify.
- Line 402: delete 'the' before 'genomic storage.
- I'd like to see some further figures relating to the consequences of gene clusters ('supergenes') co-evolving, as described towards the end of the manuscript. First, it is described on page 20 how 'diversity begets diversity' but no figures are present that show this effect. Could the authors include a plot of how diversity increases over time (e.g. showing mean diversity per locus)? Second, the mechanism of supergene formation via sequential gene additions is not clear. Could the authors please include more data and graphs on this process? I recommend plotting data on the frequency of each new 'supergene' as it appears (or just the frequency of the largest supergene), along with the average recombination rate between target loci to demonstrate how complete linkage evolves.
- On a similar note, it is unclear to me how target loci are added in the 'sequential addition' model. How much time elapses before new target loci are added? Please clarify this point in the methods section.
- I'm happy that the authors agree to make their code publically available. Could they please specify where (e.g. as supplementary material, or on a depository such as Dryad or GitHub)?

Reviewer #2 (Remarks to the Author):

I appreciate that the authors addressed my criticisms related to the biological plausibility of the results on the evolution of recombination due to genomic storage. As pointed out by the authors, there is some precedent for large-scale selection coefficients and wild fluctuations in allele frequencies in a periodic fashion. I think this greatly strengthens the papers

connection with ongoing empirical work. I particularly appreciated the predictions of what kinds of patterns would be expected in natural populations and the deeper discussion of formation of super genes by transposition.

One thing that struck me on re-reading the manuscript is that the authors make the claim that buffering via plasticity is necessary to maintain diversity in a finite population in a fluctuating selection model (e.g. lines 88-90). However, it's not obvious to me why this model will do any better than any two locus model at maintaining diversity in the face of genetic drift. It seems that if there are two loci interacting epistatically such that there's incentive to take alleles apart (indeed, the model here is essentially just one point in the space of all possible epistatic interactions) should produce similar effects. It feels like the authors need to comment on why this specific mode of epistasis is going to work when more general forms of epistasis will not. At the least, it seems like it would be interest for the authors to explore how much buffering is necessary to get stable oscillations in a finite population, since almost everything done in the manuscript assumes $p = 1$. Is 1% buffering enough? What about 10%?

I have some additional comments:

1) I still don't like the phrase on line 43 that "several lines of empirical evidence support genetic control of recombination." Obviously there is genetic control of recombination, it's done by proteins that are made by genes. I really think the authors should specifically say something along the lines of "there is heritable and selectable *variation* in recombination". I realize this is a nitpick but I think being very clear about when we talk about *genes* for something vs. *variation* for something is crucial.

2) Line 128 typo: "in not" -> "is not"

3) Line 132 typo: "locust" -> "locus"

I prefer to sign my reviews. My name is Joshua Schraiber.

Reviewer #3 (Remarks to the Author):

The authors have done a good job addressing the issues raised by myself and the other reviewers. In particular, I was happy to see that fluctuating epistasis has the mechanistic basis driving recombination rate evolution in the model is now fully integrated in the paper. With regards to novelty, I can see the authors' point and agree that there is a lot in the paper that is interesting and new, but still think a more specialised journal might be more appropriate. A few more minor comments are listed below.

l. 270: "which is more fit than all other rates": I think this phrase is problematic because recombination rates have no fitness (as the authors also nicely explain elsewhere). I would suggest to replace this phrase with a precise definition of the ES concept. The same problem arises in Table 1.

- l.48: "where desirable allelic combinations are underrepresented": I don't think this very clear. In a simple two-locus scenario without sign epistasis, negative LD means the fittest and the least fit genotypes are underrepresented, so only one of them is "desirable" even though the entire situation is desirable in the sense that selection is more efficient than without LD. Perhaps replace by something like "where genotypes with extreme fitness effects are underrepresented"?

-l.124: typo, should be "phenotypic"

- l.128: typo, "That phenotypic plasticity IS not ..."

- l.135: "experience seasonally" → "experience seasonality"?

Point-by-point response to reviews:

Reviewer #1 (Remarks to the Author):

I'd like to thank the authors for making the requested corrections to the manuscript. The mechanisms of the 'genotype storage' effect have now been elucidated and are now much clearer to understand. I only have a few additional comments.

Many thanks for your careful second review and additional useful suggestions.

- Line 148: Delete 'and' after the semicolon.

Corrected.

- Line 290: Write 'a' shorter C, rather than 'the' shorter C.

Corrected.

- I like the new text explaining the interaction between epistasis E and linkage disequilibrium D. Yet I feel it could be refined somewhat. First, on line 317 it states that "LD will be selected against". Does it make sense for LD to be 'selected against'? I take it that you mean that $ED < 0$ will favour conditions to reduce it (increased recombination in this case). Second, would you consider expanding on Figure S1 to plot how time LD mismatches with epistasis, for specific parameter spaces? Including this measure would make it clearer when recombination would be favoured, which is not entirely clear from Figure S1.

Thank you for suggesting this, we have changed Figure S1 to make this point more clear. We have added an addendum to the figure, which in detail shows how epistasis, LD, and the conditions for selection for recombination vary over a cycle of fitness oscillations for a sample trajectory. We have edited the caption to highlight when exactly selection for recombination arises. Per your suggestion, we have also made edits to the corresponding section in the main text (page 15).

- Line 325: Write "evident *that a* recombination modifier..."

Corrected.

- Lines 329–337: Did you plot the relative fitness values on Figure 2, as indicated by the colours? I don't think you did, but it not clear from the text as written. Please clarify.

The referee is correct: Figure 2 does not indicate relative fitness values of one recombination rate versus another. Instead, the purpose of Figure 2 is to report the results of local stability analyses: for any given pair of recombination rates, does the Jacobian indicate that one rate will fix over the other (blue) or that both rates will co-exist (orange). However, the results of stability analysis perfectly agree with the results of relative fitness.

We have revised the main text to explain Figure 2 and how it relates to relative fitnesses [lines 334 -345]. In particular, the values of relative fitness for one rate versus another coincide with the predictions of the local stability analysis. However, we cannot feasibly report these values, because they change dynamically during the approach to the three-locus stable attractor. The main text nonetheless highlights the important point: the relative fitness values during the approach to equilibrium correspond to the predictions of the local stability analysis – e.g., when one rate is predicted to fix, it is observed to have uniformly greater relative fitness throughout the approach towards fixation.

- Line 402: delete 'the' before 'genomic storage'.

Corrected.

- I'd like to see some further figures relating to the consequences of gene clusters ('supergenes') co-evolving, as described towards the end of the manuscript. First, it is described on page 20 how 'diversity begets diversity' but no figures are present that show this effect. Could the authors include a plot of how diversity increases over time (e.g. showing mean diversity per locus)? Second, the mechanism of supergene formation via sequential gene additions is not clear. Could the authors please include more data and graphs on this process? I recommend plotting data on the frequency of each new 'supergene' as it appears (or just the frequency of the largest supergene), along with the average recombination rate between target loci to demonstrate how complete linkage evolves.

Thank you for pointing this out and drawing our attention to the "Diversity begets diversity" section. We agree, the diversity-promoting effect of aligned clusters is rather complex. We find that a locus is more likely to be polymorphic on polymorphic genetic background than otherwise. However, we now realize that this could be misleading, because a multi-locus effect does not always translate to elevated heterozygosity compared to a single-locus model, in the absence of other mutating loci. When heterozygosity is stably maintained mutations in the cluster could break apart co-adapted complexes and reduce diversity compared to a single locus case. But, for example, for $p=1/2$ we find that this effect translates to higher heterozygosity compared to the single-locus case. Clarifying these details, which involve two counter-poised effects, would require additional text when, in fact, we are asked to reduce length; and so we have decided simply to remove these subtle points from the manuscript.

Nonetheless, to clarify sequential supergene formation we have added a new figure to the supplement and discussed it in the text (lns. 468 – 480). The Supplementary Figure S5 shows the stationary distribution of the recombination rate between the initial cluster and the newly polymorphic locus, for up to 8 loci under the sequential model of supergene formation. We studied supergene formation in a series of sequential simulations starting with an existing cluster in equilibrium and introducing polymorphism at one additional target locus. This is what we mean by the “sequential addition” model. It is a model of sequential supergene growth assuming that new genes arise after equilibrium is reached within an existing cluster. This model would apply when new mutations affecting the seasonal trait are rare, which is a reasonable assumption.

- On a similar note, it is unclear to me how target loci are added in the ‘sequential addition’ model. How much time elapses before new target loci are added? Please clarify this point in the methods section.

We have now clarified the approach to the simulation of the sequential model of supergene formation in the Methods section (lns. 701 – 710).

- I’m happy that the authors agree to make their code publically available. Could they please specify where (e.g. as supplementary material, or on a depository such as Dryad or GitHub)?

We will publish a package with frequency and individually based C programs used in the manuscript on GitHub.

Reviewer #2 (Remarks to the Author):

I appreciate that the authors addressed my criticisms related to the biological plausibility of the results on the evolution of recombination due to genomic storage. As pointed out by the authors, there is some precedent for large-scale selection coefficients and wild fluctuations in allele frequencies in a periodic fashion. I think this greatly strengthens the paper's connection with ongoing empirical work. I particularly appreciated the predictions of what kinds of patterns would be expected in natural populations and the deeper discussion of formation of super genes by transposition.

Thank you for helping us improve our manuscript, and for reading our manuscript carefully and providing suggestions, again.

One thing that struck me on re-reading the manuscript is that the authors make the claim that buffering via plasticity is necessary to maintain diversity in a finite population in a fluctuating selection model (e.g. lines 88-90). However, it's not obvious to me why this model will do any better than any two locus model at maintaining diversity in the face of genetic drift. It seems that if there are two loci interacting epistatically such that there's incentive to take alleles apart (indeed, the model here is essentially just one point in the space of all possible epistatic interactions) should produce similar effects. It feels like the authors need to comment on why this specific mode of epistasis is going to work when more general forms of epistasis will not. At the least, it seems like it would be interesting for the authors to explore how much buffering is necessary to get stable oscillations in a finite population, since almost everything done in the manuscript assumes $p = 1$. Is 1% buffering enough? What about 10%?

Thank you for pointing out this ambiguity. We have now clarified lines in question [previous lns. 88 – 90, now lns. 83 – 89]. We expanded on model description to emphasize that the genomic storage effect is a balancing selection mechanism that could also arise under similar models of epistasis in nature, such as under robustness modifiers (de Visser et al., 2003; lns. 521 – 528). And so the study has implications beyond phenotypic plasticity, which was chosen as a natural mechanism of genomic storage that is well-established and widespread in periodic environments.

Finally, as per your suggestion, we address the effect of buffering level (p) on evolution of recombination. In the deterministic limit of an infinite population we know that cycling LD (and evolution of recombination) is guaranteed for any $p > 0$. In finite populations, however, drift can disrupt genomic storage, especially when Ns_{\max} is small. Nonetheless, Figure S3 shows that recombination readily evolves across a wide range of plasticity effect sizes (e.g. $p \geq 0.25$) in an example with

populations of size $N = 10^6$ with $s_{\max} = 0.05$. This has been clarified in the main text [lns . 394 – 399].

I have some additional comments:

1) I still don't like the phrase on line 43 that "several lines of empirical evidence support genetic control of recombination." Obviously there is genetic control of recombination, it's done by proteins that are made by genes. I really think the authors should specifically say something along the lines of "there is heritable and selectable *variation* in recombination". I realize this is a nitpick but I think being very clear about when we talk about *genes* for something vs. *variation* for something is crucial.

We misunderstood your initial suggestion, and we have now edited the sentence accordingly.

2) Line 128 typo: "in not" -> "is not"

Corrected.

3) Line 132 typo: "locust" -> "locus"

Corrected.

Reviewer #3 (Remarks to the Author):

The authors have done a good job addressing the issues raised by myself and the other reviewers. In particular, I was happy to see that fluctuating epistasis has the mechanistic basis driving recombination rate evolution in the model is now fully integrated in the paper. With regards to novelty, I can see the authors' point and agree that there is a lot in the paper that is interesting and new, but still think a more specialised journal might be more appropriate. A few more minor comments are listed below.

We very much appreciated your insightful comments, as they helped us improve the manuscript and highlight which aspects of our study are unique. Thank you for giving our manuscript careful reading and useful suggestions, once again.

l. 270: "which is more fit than all other rates": I think this phrase is problematic because recombination rates have no fitness (as the authors also nicely explain elsewhere). I would suggest to replace this phrase with a precise definition of the ES concept. The same problem arises in Table 1.

Corrected.

- l.48: "where desirable allelic combinations are underrepresented": I don't think this very clear. In a simple two-locus scenario without sign epistasis, negative LD means the fittest and the least fit genotypes are underrepresented, so only one of them is "desirable" even though the entire situation is desirable in the sense that selection is more efficient than without LD. Perhaps replace by something like "where genotypes with extreme fitness effects are underrepresented"?

Corrected.

-l.124: typo, should be "phenotypic"

Corrected.

- l.128: typo, "That phenotypic plasticity IS not ..."

Corrected.

- l.135: "experience seasonally" —> "experience seasonality"?

Corrected.

REVIEWERS' COMMENTS:

Reviewer #1 (Remarks to the Author):

I'd like to thank the authors for making changes; I especially appreciate that there is now a detailed explanation on the interaction between epistasis and linkage disequilibrium on page 15. The manuscript is now in extremely good shape; I've only some minor suggestions for changes.

Line 37: Write "subject *of* longstanding interest".

Line 55: Delete 'the' before 'drift'.

Line 270–271: Rewrite to read "at *both the* plasticity...".

Line 274–277: I think the opening sentence for the paragraph should be nuanced. As currently written, it implies that non-zero recombination rates evolves over most of the parameter space for all C . Yet for $C = 80$ for example, non-zero recombination only evolves if there is initially zero recombination (and even then, the non-zero rate does not go to fixation). Furthermore, it seems better to write 'non-zero' recombination instead of 'positive' recombination in line 274 (as the latter implies 'negative' recombination can exist!).

Line 474: Correct spelling of 'mating'.

Figure S1: I think there was a misunderstanding of my suggestion as to how to update Figure S1 (my apologies for the ambiguity in my previous review). I had it in mind that the authors could add panels complementing the third row, which plot to what extent there is a mismatch between epistasis and linkage disequilibrium. For example, with time on the X-axis, and cumulative time where $ED < 0$ on the Y-axis. Given Figure S1 is already rather busy, I'll leave it to the authors to decide whether it'll be worth including this extra information.

Reviewer #2 (Remarks to the Author):

I am happy with the paper, and the authors have addressed my major concerns. I think the paper presents an interesting and plausible path toward both the evolution of recombination in the first place and the evolution of super genes.

I prefer to sign my reviews. My name is Joshua Schraiber.

Point-by point response to reviewers:

Reviewer #1 (Remarks to the Author):

I'd like to thank the authors for making changes; I especially appreciate that there is now a detailed explanation on the interaction between epistasis and linkage disequilibrium on page 15. The manuscript is now in extremely good shape; I've only some minor suggestions for changes.

Once again, thank you for making valuable suggestions that helped to immensely improve our MS.

Line 37: Write "subject *of* longstanding interest".
Corrected.

Line 55: Delete 'the' before 'drift'.
Corrected.

Line 270–271: Rewrite to read "at *both the* plasticity...".
Corrected.

Line 274–277: I think the opening sentence for the paragraph should be nuanced. As currently written, it implies that non-zero recombination rates evolves over most of the parameter space for all C . Yet for $C = 80$ for example, non-zero recombination only evolves if there is initially zero recombination (and even then, the non-zero rate does not go to fixation). Furthermore, it seems better to write 'non-zero' recombination instead of 'positive' recombination in line 274 (as the latter implies 'negative' recombination can exist!).

Thank you for pointing this out. We meant to refer to the ES recombination rate, and have corrected the text accordingly.

Line 474: Correct spelling of 'mating'.
Corrected.

Figure S1: I think there was a misunderstanding of my suggestion as to how to update Figure S1 (my apologies for the ambiguity in my previous review). I had it in mind that the authors could add panels complementing the third row, which plot to what extent there is a mismatch between epistasis and linkage disequilibrium. For example, with time on the X-axis, and cumulative time where $ED < 0$ on the Y-axis. Given Figure S1 is already rather busy, I'll leave it to the authors to decide whether it'll be worth including this extra information.

We did consider such a complementary panel to the third row and found that it is also quite busy. As you noted, the figure is already very busy and we choose not to

make further changes on it.

Reviewer #2 (Remarks to the Author):

I am happy with the paper, and the authors have addressed my major concerns. I think the paper presents an interesting and plausible path toward both the evolution of recombination in the first place and the evolution of super genes.

I prefer to sign my reviews. My name is Joshua Schraiber.

We are very thankful to prof. Schraiber for all of his suggestions and multiple careful readings of our manuscript.